# Transport of Biomass Burning Aerosol into the Extratropical Tropopause Region over Europe via Warm Conveyor Belt Uplift

Philipp Joppe<sup>1,2</sup>, Johannes Schneider<sup>2</sup>, Jonas Wilsch<sup>2</sup>, Heiko Bozem<sup>1</sup>, Anna Breuninger<sup>5</sup>, Joachim Curtius<sup>5</sup>, Martin Ebert<sup>7</sup>, Nicolas Emig<sup>1</sup>, Peter Hoor<sup>1</sup>, Sadath Ismayil<sup>7</sup>, Konrad Kandler<sup>7</sup>, Daniel Kunkel<sup>1</sup>, Isabel Kurth<sup>1</sup>, Hans-Christoph Lachnitt<sup>1</sup>, Yun Li<sup>3</sup>, Annette Miltenberger<sup>1</sup>, Sarah Richter<sup>5</sup>, Christian Rolf<sup>4</sup>, Lisa Schneider<sup>7</sup>, Cornelis Schwenk<sup>1</sup>, Nicole Spelten<sup>4</sup>, Alexander L. Vogel<sup>5</sup>, Yafang Cheng<sup>6</sup>, and Stephan Borrmann<sup>1,2</sup>

**Correspondence:** Philipp Joppe (phjoppe@uni-mainz.de)

**Abstract.** Aerosol particles in the extratropical upper troposphere and lower stratosphere (exUTLS) play a crucial role for the Earth's radiative budget. High temporal and spatial resolution measurements in the exUTLS are important to study mixing processes and their climate impact. Here, we present measurements from the TPEx mission (Tropopause composition gradients and mixing Experiment) an aircraft mission in June 2024 over Europe. The measurement platform, a Learjet 35A, was equipped with in-situ trace gas and aerosol measurements and filter samplers for offline analysis. For vertical gradient measurements of trace species and aerosol, we conducted redundant measurements on a fully automated towed sensor shuttle (TOSS) 200 m below the aircraft.

On 17 June 2024, we observed a streamer with elevated aerosol number concentration of up to 800 particles per cm<sup>3</sup> between 100 nm and 1  $\mu$ m. This is higher by a factor of more than two, up to four, compared to the UTLS, respective tropospheric, background. Carbon monoxide (CO) mixing ratios were larger than 100 ppbv. Backward trajectories indicate that this pollution is transported from Canadian wildfires in the lower troposphere towards Europe, where it was uplifted on the edge of a warm conveyor belt into the tropopause region. There mixing with chemically stratospheric air occurred. The TOSS measurements also allow the calculation of the potential temperature gradient ( $\Delta\theta \cdot \Delta z^{-1}$ ). We observed a change towards smaller gradients within the region of the polluted air masses, which is presumably due to an increase of  $\theta$  at lower altitudes by radiative heating

as a consequence of the transported refractory black carbon.

Copyright statement. TEXT

<sup>&</sup>lt;sup>1</sup>Institute for Atmospheric Physics, Johannes Gutenberg University Mainz, Mainz, Germany

<sup>&</sup>lt;sup>2</sup>Particle Chemistry Department, Max Planck Institute for Chemistry, Mainz, Germany

<sup>&</sup>lt;sup>3</sup>ICE-3: Troposphere, Forschungszentrum Jülich, Jülich, Germany

<sup>&</sup>lt;sup>4</sup>ICE-4: Stratosphere, Forschungszentrum Jülich, Jülich, Germany

<sup>&</sup>lt;sup>5</sup>Institute for Atmospheric and Environmental Sciences, Goethe University Frankfurt, Frankfurt am Main, Germany

<sup>&</sup>lt;sup>6</sup>Aerosol Chemistry Department, Max Planck Institute for Chemistry, Mainz, Germany

<sup>&</sup>lt;sup>7</sup>Institute for Applied Geoscience, Technical University Darmstadt, Darmstadt, Germany

#### 1 Introduction

The chemical composition of the upper troposphere and lower stratosphere (UTLS) is of high importance for the Earth's radiative budget (Kremser et al., 2016). Changes in the composition of this region have a large impact on the Earth's surface temperature, due to the radiative sensitivity of the surface temperature with respect to changes in trace gases such as ozone (O<sub>3</sub>), water vapor (H<sub>2</sub>O), and methane (CH<sub>4</sub>) (Riese et al., 2012). The total aerosol effective radiative forcing (AERF) is highly variable and depends on the chemical composition of the aerosol particles. For example, at the top of the atmosphere (TOA), the global radiative effect of sulfate aerosol is a cooling effect up to about -1.3 Wm<sup>-2</sup> whereas the global radiative effect of black carbon (BC) shows a strong heating effect of up to 0.9 Wm<sup>-2</sup> (Masson-Delmotte et al., 2023; Kalisoras et al., 2024; Ramanathan and Carmichael, 2008; Räisänen et al., 2022). The aerosol chemical composition in the UTLS is influenced by different processes: In the tropical regions, aerosol particles and precursor gases are transported from the planetary boundary layer (PBL) into the tropical transition layer (TTL) by deep convection on a timescale of minutes to hours (Froyd et al., 2009; Fueglistaler et al., 2009; von Hobe et al., 2021). Such deep convective transport can be associated with new particle formation events, due to the presence of freshly oxidized volatile organic compounds (VOCs) and the nucleation of these oxidation products in the convective outflow at high altitudes (Curtius et al., 2024). The newly formed particles in the UTLS are redistributed towards the extratropical and polar regions under the influence of the Brewer-Dobson circulation (BDC) (Andersson et al., 2015; Kremser et al., 2016).

The chemical composition of the lowermost stratosphere (LMS) is influenced by the shallow branch of the BDC, where the meridional transport from tropical to polar latitudes takes from months up to more than one year (Ploeger et al., 2021). There are several additional processes which influence the chemical composition and other properties of the aerosol on shorter timescales and more locally, such as convective events, planetary and synoptic scale waves, associated with baroclinic instabilities and vertical transport from the PBL to the UT ahead the surface cold fronts by warm conveyor belts (WCBs). These processes often generate strong shear, thus favorable conditions for turbulence and mixing (Zahn et al., 2000; Brioude et al., 2007; Kaluza et al., 2021, 2022; Lachnitt et al., 2023).

40

50

A WCB is part of an extratropical cyclone, which is characterized by the transport of moist air masses over large horizontal distances in combination with a strong ascent into the UT (Harrold, 1973; Heitmann et al., 2024). WCBs can be described from a Lagrangian perspective as a compact trajectory bundle in the vicinity of a cyclone with special characteristics, such as strong ascent and drying of air masses (Heitmann et al., 2024; Schwenk and Miltenberger, 2024). WCBs are capable to transport trace gases, aerosol precursor gases, and aerosol particles from the PBL into the UTLS. Furthermore, WCB transport is able to enhance cross-tropopause mixing by triggering gravity waves, enhancing shear and radiative heating as a consequence of cloud formation. Therefore WCBs can affect the chemical composition of the UTLS (Cooper et al., 2004). Recent studies of WCBs focused on the transport of water vapor (H<sub>2</sub>O), mineral dust, and the precursor gas sulfur dioxide (SO<sub>2</sub>) into the tropopause region (Madonna et al., 2014; Marelle et al., 2015; Fromm et al., 2016; Schwenk and Miltenberger, 2024).

The chemical composition of stratospheric aerosol particles is influenced by tropospheric compounds that are transported to

the UTLS. Tropospheric sulfur compounds, including anthropogenic and volcanic SO<sub>2</sub> emissions, contribute to the abundance of sulfate in the stratospheric aerosol (Andersson et al., 2015; Joppe et al., 2024). In addition to sulfate, stratospheric aerosol particles also contain carbonaceous compounds, such as organic carbon (OC; or organic aerosol) and black carbon (BC). One major source for OC and BC in the UTLS is biomass burning (BB) (Murphy et al., 2007; Schwarz et al., 2008; Ditas et al., 2018; Ma et al., 2024).

Organic aerosol (OA) can be divided into two types of aerosol, primary organic aerosol (POA) which is directly emitted by the source (e.g., pollen, spores, sea salt and its coatings or biomass burning) and secondary organic aerosol (SOA) which is formed by gas-to-particle conversion out of precursor gases. During summer months, biomass burning organic aerosol (BBOA) contributes up to 16 % to the total OA over Europe while up to 70 % of this BBOA are formed secondarily (Theodoritsi and Pandis, 2019). Additionally, the BB aerosol contributes on average 10 %, sometimes up to 50%, to the aerosol number concentration for particle sizes between 0.25 and 2 µm in the northern LMS (Kremser et al., 2016). The transport of these aerosol particles into the LMS has been investigated for fast processes, like convective uplift and contributions by pyroconvection (Fromm et al., 2010; Yu et al., 2019; Peterson et al., 2021; Ma et al., 2024).

In this study we show that aerosol particles from small wildfires without large smoke plumes can be transported into the UTLS by WCB uplift and subsequent cross-tropopause mixing far away from the original source. Furthermore, we assess the effect of the observed small-scale pollution streamers on the static stability in the tropopause region. For this, we use in-situ measurement data in combination with transport diagnostics based on LAGRANTO back trajectories and model data from the TPEx (tropopause composition gradients and mixing experiment) mission, conducted over Germany and northern Europe in June 2024.

#### 2 Methods

70

#### 2.1 TPEx flight F07

The TPEx aircraft campaign was conducted over Germany and northern Europe for three weeks in June (03 - 21 June) 2024. The campaign base was at Hohn airfield ( $54^{\circ}18'49''N$ ,  $9^{\circ}32'17''E$ ), near Rendsburg, Germany. The research aircraft was a Learjet 35A, owned and operated by the Gesellschaft für Flugzieldarstellung (GFD), equipped with several online and offline measurement instruments inside the cabin and an underwing pod outside the cabin. In total, we conducted nine research flights reaching from the boundary layer up to 12 km. In this study we focus on research flight F07, which took place on 17 June 2024. For this flight we used the capability of the Learjet to tow a second measurement platform (The TropoPause Composition towed sensor shuttle; TPC-TOSS (Frey et al., 2009; Finger et al., 2016; Klingebiel et al., 2017; Bozem et al., 2025) on a steel cable below the aircraft. The TPC-TOSS and the aircraft were equipped with identical instrumentation for gradient measurements of temperature, ozone and aerosol number concentration between 100 nm and 1  $\mu$ m. The goal of research flight F07 was to probe a region with a variable tropopause altitude (see Fig. 1a). As consequence we expected enhanced cross-tropopause mixing as a consequence of a low pressure system over the North Sea west of Norway and predicted low Richardson numbers in the restricted air space (not shown). The vertical cross-section along the flight path (Fig. 1b) indicates some predicted variability at

and above the 2 PVU tropopause as well as some stratospheric intrusions which are growing in spatial extent during the flight (green patches in Fig. 1b). Furthermore, the forecast for WCB outflow indicated the potential for aged WCB outflow in the area of interest (see Fig. 2).

**Figure 1.** ECWMF forecast data for 17 June 2024, 06 UTC (F07). The potential temperature along the thermal WMO tropopause and the flight path of F07 (red solid line) are shown in panel (a). Panel (b) shows a vertical cross-section of potential vorticity (PV) along the flightpath (red solid) line with the 2 PVU surface as marker for the dynamical tropopause in the model (white solid line).

**Figure 2.** ICON-EU forecast data for the average WCB outflow age and the bottom altitude of the WCB outflow on basis of the forecast from 16.06.2024 00:00 UTC, valid for 17.06.2024 06:00 UTC.

#### 2.2 Instrumentation

100

105

## 2.2.1 In-situ aerosol measurements

The size distribution and number concentration of aerosol particles in the accumulation mode between 100 and 1000 nm, were measured using two ultra-high sensitivity aerosol spectrometers (UHSAS, Droplet Measurement Technologies) (Cai et al., 2008; Kupc et al., 2018; Mahnke et al., 2021), one in the cabin of the Learjet and one in the TPC-TOSS (Bozem et al., 2025). Both instruments deliver measurement data with a resolution of 1 Hz. The measuring principle is based on light scattering in the infrared spectral range. The UHSAS uses a Nd<sup>3+</sup>:YLiF<sub>4</sub> solid state laser with an operating wavelength of 1054 nm (Cai et al., 2008; Kupc et al., 2018). The laser mode has an intracavity power of approximately 1.1 kW · cm<sup>-2</sup> and is perpendicular to the particle stream. Aerosol particles are actively pumped into the detection unit through a jet assembly with a sample flow of 50 cm<sup>3</sup>· min<sup>-1</sup> and are focused to a narrow particle beam with a sheath flow.

The UHSAS in the cabin of the Learjet has been modified and rebuilt to a new housing for the use onboard a research aircraft. It is connected to the aerosol sampling inlet which consists of stainless steel and has a tip diameter of 1.55 mm. The inlet expands to 30 mm before it enters the cabin. From the cabin-side of the inlet, in total five 0.25 inch tubes are embedded into the large tube to connect the individual aerosol instruments with the inlet system. The flow was controlled by a software with input parameters for different aircraft speeds and altitudes aiming to sample close to isokinetic conditions. In the measured size range of the UHSAS we calculated transmission efficiencies of 86 % at the boundaries and 95 % at diameters around 300 nm. These calculations were performed for an ambient pressure of 300 hPa and 240 K using the Particle Loss Calculator by von der Weiden et al. (2009).

The UHSAS inside the TPC-TOSS is the special version of the instrument for airborne measurements (model UHSAS-A), manufactured for the use as an underwing probe. For previous operation at altitudes up to 21 km, the original pump had been replaced and an internal computer had been added for internal data recording and saving (Mahnke et al., 2021). The flow systems differ slightly between both instruments: The UHSAS-A in the TPC-TOSS is equipped with a second mass flow

10 controller (MFC) to also control the sheath flow whereas the cabin instrument uses only one MFC for the sample flow, and the sheath flow is supplied by the remaining air flow. Before the deployment during TPEx, we performed several characterization and calibration measurements to ensure the comparability of the measurements. These measurements and characterizations are described in Bozem et al. (2025).

For larger aerosol particles, we operated an optical particle counter (OPC, model 11-S, GRIMM) in the cabin of the Learjet to measure the aerosol size distribution of particles larger than 250 nm. Due to particle losses in the inlet system the OPC measurements yield particle size distributions up to around  $10 \mu m$ . The particle loss calculation was done using the Particle Loss Calculator described in von der Weiden et al. (2009). From these OPC we get information of the size distribution every 6 s.

The aerosol number concentration for nucleation and Aitken-mode particles was measured in the cabin of the Learjet by a multi-channel condensation particle counter (mc-CPC, consisting of 3 Grimm Aerosol Technik model 5410-Sky CPCs). The individual CPCs are operated with Fluorinert (FC-43 3M<sup>TM</sup>) as working fluid and are set to different cut-off diameters in order to obtain information on new particle formation events and the particle growth. With the current configuration, we achieved cut-off diameters of 12 nm and 16 nm, inferred from laboratory calibrations. Here, we decided to operate two of the three mc-CPC channels at the same cutoffs, to cross-check the data quality of the aerosol number concentration during the flights.

The chemical composition of non-refractory aerosol particles between 50 and 800 nm was measured using an aerosol mass spectrometer in the cabin of the Learjet. This instrument, which is based on the miniAMS by Aerodyne Research Inc., had been designed for operation in the IAGOS-CARIBIC (In-service Aircraft for a Global Observing System - Civil Aircraft for the Regular Investigation of the Atmosphere Based on an Instrument Container) project (CARIBIC-AMS). It is operated with a time resolution of 30 s resulting in a horizontal resolution of around 5 km in the UTLS. The CARIBIC-AMS is thus comparable to other AMS instruments. Although it is designed to be operated fully automated during IAGOS-CARIBIC flights, we operated the CARIBIC-AMS manually during the TPEx mission (Schneider et al., 2025).

## 2.2.2 In-situ trace gas measurements

115

125

For the simultaneous measurement of the trace gases nitrous oxide  $(N_2O)$  and carbon monoxide (CO) the Quantum Cascade Laser based spectrometer University Mainz airborne QCL Spectrometer (UMAQS) is used (Müller et al., 2015; Kunkel et al., 2019). It is based on the Aerodyne Research Inc. Quantum Cascade Laser Mini Monitor which uses an astigmatic multi path Herriot cell with an optical pathlength of 76 m operated at a cell pressure of 53 hPa. The measurement principle is based on infrared absorption spectroscopy at characteristic absorption lines of  $N_2O$  and CO in the 2200 cm<sup>-1</sup> range. For operating the instrument on airborne platforms, the pressure within the measurement cell is controlled at 53 hPa. The instrument is calibrated in-situ to account for instrument drifts. Data are obtained at a time resolution of 1 Hz, finally limited by the gas exchange time of the measurement cell. This allows to measure atmospheric concentrations of  $N_2O$  with a noise level of 0.08 ppbv  $(2\sigma)$  and a reproducibility of 0.2 ppbv  $(2\sigma)$ . For CO measurements the noise level amounts to 0.38 ppbv  $(2\sigma)$  and for the reproducibility we reach 0.7 ppbv  $(2\sigma)$  (Müller et al., 2015; Kunkel et al., 2019).

Ozone was measured using two modified 2BTech Model 205 instruments (Johnson et al., 2014; Bozem et al., 2025). One O<sub>3</sub>

instrument was mounted in the wing pod of the Learjet and the second one was operated in TPC-TOSS. The measuring prin-45 ciple is based on UV absorption at the wavelength of 254 nm at ambient pressure. The time resolution of these measurements is 2 seconds with an uncertainty of 0.5 % + 2.0 ppbv.

Temperature and humidity data on the Learjet and the TPC-TOSS are measured by MCH (MOZAIC capacacitive hygrometer sensors, which consist of a capacitive relative humidity sensor by the company Vaisala, Finland and a PT100 resistance sensor for the temperature measurements (Helten et al., 1998; Smit et al., 2014). These sensors have been in regular service during then the MOZAIC program and now the IAGOS research infrastructure (Petzold et al., 2015). The MCHs were calibrated against a dew point hygrometer (MBW373) before and after the campaign in the atmospheric simulation chamber at Jülich (Smit et al., 2014). Based on the calibration, the MCHs give an uncertainty of 5 % relative humidity with respect to liquid water in the upper troposphere, tropopause and lowermost stratosphere and 0.5 K for temperature (Smit et al., 2014)).

The Fast In-situ Stratospheric Hygrometer (FISH) provides reliable water vapor measurements with the Lyman-α photofragment fluorescence technique aboard research aircraft for almost 30 years (Zöger et al., 1999; Meyer et al., 2015). The uncertainty estimation based on our regular calibrations during the TPEx campaign is 4.9 % of the respective measured value plus a constant uncertainty of 0.77 ppmv.

## 2.2.3 Collection of filter and impactor samples for the offline analysis

150

In addition to the in-situ measurements, we operated an in-house developed and manufactured filter sampler to gain information on the chemical composition of organic aerosol particles (SOAP) (Breuninger et al., 2025). Here, the collection of up to five different filter samples (47 mm diameter, Whatman<sup>TM</sup> QM-A, cytiva) was possible due to a system of switchable magnetic valves and a bypass. The filters were sampled with 60 L· min<sup>-1</sup> at standard conditions. After sampling, the filters were sealed in aluminium foil and stored in a freezing box to minimize artefacts and losses of the collected aerosol particles (Resch et al., 2023).

The filter analysis was carried out by extracting the filters in either a mixture of 90 % ultrapure water and 10 % methanol or pure methanol. After cutting pieces of the filter and adding the solvent, the vials are placed on an orbital shaker (KS-15, Edmund Bühler GmbH) at 300 rpm. After extraction, the extract is filtered through a disposable polytetrafluoroethylene (PTFE) filter (pore size: 0.2 μm, macherey nagel). The extracts were measured by using ultra high-pressure liquid chromatography (Vanquish Flex) coupled with high-resolution orbitrap mass spectrometry (Q Exactive Focus Hybrid Quadrupole Orbitrap, both Thermo Fisher Scientific). The separation was carried out according to previous studies from (Ma et al., 2022; Ungeheuer et al., 2021; Thoma et al., 2022), on a C<sub>18</sub>-Column (CORTECS<sup>TM</sup> T3, 2.7 μm x 150 mm, Waters), using a gradient elution with ultrapure water and methanol. The peak identification and analysis was done using the software FreeStyle<sup>TM</sup> 1.8 SP2 (Thermo Fisher Scientific).

Additionally, during all flights UTLS particle samples were collected by the miniaturized MultiMINI8 casacade impactor unit.

This self-developed Integrated Aerosol Sampling System, which is based on the former MultiMINI design (Ebert et al., 2016) was designed for the use within the wing pod of the Learjet. In total 8 two stage impactors (particle diameter: fine stage 0.1 – 1 µm; coarse stage > 1 µm) are integrated in this sampling unit. Particles were deposited on TEM grids, which are best suited for

later offline individual particle analysis by electron microscopic methods. Size, morphology, and elemental composition of the particles were studied by scanning electron microscopy (SEM) and energy-dispersive X-ray microanalysis (EDX). SEM-EDX was carried out with a FEI ESEM Quanta 400F (Eindhoven, The Netherlands) equipped with a X max 80 energy-dispersive X-ray detector (Oxford Instruments, Abingdon, UK), which enables the analysis of elements with  $Z \ge 5$ . All investigations were carried out at 12.5 kV acceleration voltage and spot size 4 (beam diameter  $\approx 30$  nm). The particles were studied without coating in the high vacuum mode of the instruments ( $\approx 5 \cdot 10^{-6}$  mbar sample chamber pressure). Particle types were classified based on chemical composition and in case of biomass burning particles and soot additionally based on their typical morphology.

#### 185 2.3 Meteorological support data and transport diagnostics

180

190

For the analysis of meteorological parameters, we used the ERA5 reanalysis data with a temporal resolution of 1h and a horizontal grid spacing of 0.3°. Additionally, on the basis of the native variables we calculated meteorological parameters such as vertical wind shear (S<sup>2</sup>), static stability (squared Brunt-Vaisala frequency, N<sup>2</sup>), potential vorticity (PV) and equivalent latitude (EQLAT) (e.g., Lary et al., 1995; Krause et al., 2018; Joppe et al., 2024). In order to gain additional information about the chemical composition of the UTLS, we used the forecast data of the Copernicus Atmosphere Monitoring Service (CAMS), which is based on the Integrated Forecasting System 48r1 (IFS) with additional chemical modules for the chemical analysis, such as aerosol, reactive gases and greenhouse gases. This forecast is available twice a day (00 and 12 UTC) with a horizontal resolution of 40 km and 137 vertical levels (e.g., Benedetti et al., 2009; Morcrette et al., 2009; Rémy et al., 2019). We interpolated the CAMS data onto the flight tracks as we did for the ERA5 data. For analyses of air mass origin, we calculated backward trajectories from the position of the Learjet with LAGRANTO (Sprenger and Wernli, 2015) based on combined ICON global operational analysis and forecasts. We used the ICON global model for the trajectories instead of ERA5 to make use of the higher horizontal resolution. ICON global analysis is available every 6 h and is combined with short-term forecast to achieve an hourly resolution of wind field data. ICON global analysis and forecast data are available on a native R03B07 grid (corresponding to about 13.5 km effective grid spacing) and has been re-gridded to a regular longitude-latitude grid with 0.15° spacing. In the vertical, ICON global data comprises 120 levels with a spacing of about 300 m between about 4 km and 13 km altitude. The trajectories are calculated 10 days back in time. In addition, we combined the backward trajectories with the CAMS Global Fire Assimilation System (GFAS) version 1.2 to check if the trajectories crossed potential biomass burning regions. GFAS assimilates fire radiative power (FRP) on basis of satellite measurements to give daily estimates for biomass burning locations (Kaiser et al., 2012; Rémy et al., 2017). The fire locations are taken from the GFAS data set with detected fire radiative power signals in cloud-free regions and reports by the Canadian Wildland Fire Information System (CWFIS, https://cwfis.cfs.nrcan.gc.ca/interactive-map; last access: 09.01.2025) in order to close the gap of the satellite retrievals in regions with clouds.

#### 2.4 Determination of the tropopause

Several definitions for the extratropical tropopause exist, such as the dynamical tropopause based on a certain threshold value of potential vorticity (PV) or the gradient of PV (Kunz et al., 2011; Turhal et al., 2024). In addition to dynamical tropopause

definitions there are also definitions based on the temperature, such as the thermal tropopause defined by the temperature lapse rate (World Meteorological Organization (WMO, 1997)) or the tropopause defined on the static stability (gradient in potential temperature  $\theta$ ) (Tinney et al., 2022). Finally, one can also define a chemical tropopause, based on trace gases, such as  $O_3$  or nitrous oxide ( $N_2O$ ) (Bethan et al., 1996; Müller et al., 2015; Joppe et al., 2024). In our study, we use the chemical tropopause based on the mixing ratio of mainly  $N_2O$  similar to Müller et al. (2015).  $N_2O$  has no sinks in the troposphere and a lifetime exceeding 100 years, so its mixing ratio remains constant throughout the troposphere, with a gradient at the tropopause, caused by the sinks in the stratosphere. Therefore, we can chemically define tropospheric and stratospheric air masses, solely based on  $N_2O$  measurements (Müller et al., 2015). The tropopause height is highly variable in time and space and depends further on the used definition. During summer months the dynamical 2 PVU tropopause tends to be lower than thermal WMO tropopause or the PV-gradient tropopause (Kunz et al., 2011; Turhal et al., 2024).

## 3 Transported biomass burning aerosol and its effects in the LMS

## 3.1 Observation of an aerosol polluted filament

In this study, we focus on the first of the five triangular patterns flown within the restricted airspace between 07:20 and 07:50 UTC (Fig. 1). The time series shows a very small-scale pollution event with an increase in particle number concentration by more than a factor of two compared to the UTLS background (Fig. 3). Furthermore, we observe an enhancement in CO of more than 20 ppbv in a region with stratospheric  $N_2O$  ( $N_2O$  < 336 ppbv) and  $O_3$  ( $O_3$  > 150 ppbv). The features we investigate in the following are also visible at higher altitudes later in the flight, but less dominant. Figure 3 shows a period of the time series for selected in-situ measurements carried out on both platforms. The measurements on the TPC-TOSS were shifted in time according to the lateral offset of the measurement platform behind the towing Learjet.

In more detail, we focus on the four segments with increased aerosol number concentration of 0.1 to 1.0 μm particles measured by the UHSAS, i.e. 07:24 - 07:26; 07:27 - 07:29; 07:42 - 07:44 and 07:45 - 07:47 UTC, which coincide with enhanced CO and N<sub>2</sub>O as well as with decreased H<sub>2</sub>O. The O<sub>3</sub> mixing ratios also decrease, but only to a very small extent from above 160 ppbv to around 150 ppbv. The trace gas mixing ratios indicate tropospherically influenced air masses with values of more than 110 ppbv CO, while the N<sub>2</sub>O clearly shows stratospherically influenced air with values around 335.5 ppbv, i.e. lower than the tropospheric background of 337 ppbv. The interpretation of the mixed air masses into chemically stratospheric air is also supported by the O<sub>3</sub> mixing ratios above 150 ppbv and H<sub>2</sub>O values near 100 ppmv H<sub>2</sub>O. From the aerosol perspective, we observe increased particle number concentrations compared to the tropospheric measurements between 07:30 and 07:40 UTC for particles between 100 nm and 1 μm.

Several processes could have caused the observed high aerosol number concentration, which we discuss in the following. More precisely, enhanced aerosol number concentration can be caused by new particle formation with a high number of very small particles in the nucleation mode. Other possibilities for high aerosol number concentration are anthropogenic or biomass burning pollution events with high particle concentrations in the Aitken- and accumulation mode. Further sources are dust events or volcanic plumes which would result in a high aerosol number concentration in the coarse mode. Especially the

mc-CPC measurements can be used to differentiate between fresh new particle formation or more aged and enlarged aerosol particles. Between 07:43 and 07:46 UTC, we observe that the size channel with particles larger than 12 nm strongly differs from the channel with particles larger 16 nm. This might be an indication for a recent new particle formation event in this airmass. However, to prove this hypothesis, further analysis is required which is beyond the scope of this study.

Figure 3. Time series of trace gas and aerosol in-situ measurements for F07 on 17 June 2024 for the period between 07:20 and 07:50 UTC. Measurements conducted on the Learjet are in darker colors whereas measurements from the TPC-TOSS are in lighter colors. Row (a) shows the potential temperature ( $\theta$ ) with the tropopause derived from vertical trace gas profiles (dashed line), (b) trace gases CO (black) and  $N_2O$  (green), (c) aerosol number concentration between 100 nm and 1  $\mu$ m, (d)  $H_2O$  (blue) and  $O_3$  (red) as well as (e) total aerosol number concentration from 12 nm (yellow) and 16 nm (blue) measurements. The orange shaded boxes mark the four analyzed periods with enhanced aerosol number concentration  $N_{UHSAS}$  and CO mixing ratio.

The presence of polluted air masses in chemically stratospheric air is also supported by the vertical profiles of the trace gases (Fig. A1), which can be used as stratospheric tracers, namely  $O_3$  and  $N_2O$ . Both trace gases show rather constant mixing ratios in the troposphere and then a stratospheric increase for  $O_3$ , and a decrease for  $N_2O$ , respectively (Müller et al., 2015; Joppe et al., 2024). From these profiles (see Fig. A1), we identify the chemical tropopause at altitudes around 308 K potential temperature. We accounted for the tropopause variability of less than 5 K by taking the mean between initial ascent and descent. The observed pollution event is located 2 to 4 K above this chemical tropopause at potential temperatures of 311 K in a layer with decreasing  $N_2O$  which is clearly stratospheric air. Compared to the in-situ measurements, the interpolated ERA5 data along the flight path is showing a higher tropopause with a thermal tropopause at 315 K and a dynamical 2 PVU tropopause

50-70 hPa above the flight path of the first pattern (see Fig. 1). This model observation fits well to radiosonde sounding in Schleswig (12 UTC) which yield a tropopause height at 9.1 km or 320 K  $\theta$  based on the definition of Tinney et al. (2022). Nevertheless, we use for our analysis the chemical tropopause based on in-situ measurements to minimize errors which might arise from the model resolution and other model deficiency.

Figure 4 shows the pattern of the first altitude stack at approximately 7400 m (FL240) after the TPC-TOSS deployment. The polluted air masses appear as a small band which is oriented north-eastwards, crossing the flight pattern. Furthermore, the different chemical characteristics of the probed air masses can be seen: While we observe clean and unpolluted tropospheric air masses in the eastern part of the pattern, the western part is characterized by stratospheric air with the polluted streamer embedded.

Figure 4. Flight track of the first complete triangular pattern with deployed TPC-TOSS within the restricted air space. The pattern is flown at FL240 (approx. 7400m) between 07:24 and 07:47 UTC. The colorcode represents different in-situ measurements onboard the Learjet: (a) total aerosol number concentration between 0.1 and 1  $\mu$ m, (b) CO, (c) N<sub>2</sub>O and (d) H<sub>2</sub>O measurements. The red box surrounds the observed polluted air masses.

Figure 4 gives a rough overview of the pollution within the streamer. We are interested in its difference in particle size distribution compared to the local and the tropospheric background. For this purpose we focus in the following on the aerosol size distribution and the chemical composition with respect to non-refractory compounds, such as particulate sulfate, ammonium, nitrate and organics. We can clearly see that the particle number concentration measured by the UHSAS during the lowest

pattern correlates with the CO mixing ratio (see Fig. 5a). This suggests that the measured aerosol is of tropospheric origin with most likely the same source as the high CO mixing ratios within the LMS.

## 3.2 Identification of biomass burning in the UTLS

Possible sources for the observed high CO values in the stratosphere are advected biomass burning residues or local uplifted pollution from the surface. We analyzed the aerosol size distributions measured by the UHSAS at the Learjet in order to find some hints for possible biomass burning in these size distributions. Therefore, we calculated the aerosol volume distribution over 10 s, which is sufficient enough to average over the full peak of elevated aerosol number concentration. Furthermore, we also calculated volume distributions during the local UTLS background between the two consecutive pollution events and the tropospheric background in the middle of the pattern without observed pollution. The detailed times are also given in Table C1. As optical particle detection reveals some uncertainties due to absorbing aerosol particles, especially when calculating volume distributions, we additionally provide an uncertainty range. For this, we use the sizing uncertainties for a UHSAS according to Moore et al. (2021). Based on in-situ observations and laboratory studies, they provide sizing errors for different particle types, especially for wildfire biomass burning aerosol. Following Moore et al. (2021), we calculated the uncertainty range for potential 3 % oversizing up to 20 % undersizing as consequence of light absorption by wildfire biomass burning aerosol particles. The inferred volume distributions (Fig. 5b) show significant differences between the UTLS background and the polluted air masses. Even though the uncertainty range is high, we observe a modal distribution with a mode between 200 and 400 nm during the pollution events. This mode is robust against the measurement uncertainty and can be differentiated from the UTLS and tropospheric background. Such a mode in the volume distribution has previously been reported for observed aged biomass burning aerosol (Alonso-Blanco et al., 2014; Ditas et al., 2018; Brock et al., 2021; Schill et al., 2022; Holanda et al., 2023).

**Figure 5.** Detailed analysis of measured polluted air masses with a time-series of particle number concentrations measured by the UHSAS on the Learjet (blue), and CO (black) (a), as well as measured volume distribution averaged over 10 s (see Table C1) for the polluted air masses (dark red), the UTLS background (bright blue) and the tropospheric background for the pattern (black). The shaded area represents the uncertainty range due to sizing errors of absorbing particles (b). The dashed lines represent the original measurement data with the color corresponding to the atmospheric regime.

Particle composition information are gained by the individual particle analysis of the MultiMINI8 casacade impactor samples. Here, we performed offline analysis by SEM-EDX. In the analyzed impactor sample collected on the first leg in the restricted air space (7:16–7:44 UTC), BB particles were the most abundant refractory particle type. Two different types of BB particles can be identified in the SEM. Firstly, carbon dominated particles with increased potassium, sulfur and in some cases chlorine contents are observed. Secondly, soot particles, characterized by their typical morphology and the very low oxygen content of the C-rich particles are detected. These chemical signatures are highly characteristic for BB particles (Li et al., 2003). Two secondary electron images with the referring energy dispersive X-ray spectra for the two types of observed BB particles are shown in Fig. 6. Beside the BB particles, also some Ca-rich particles, Fe-rich particles and aluminosilicate (soil) particles were observed. The second used offline sampling method by the SOAP does only show very weak signals signals of BB influence for this flight. The results are given in appendix D.

**Figure 6.** Secondary electron images and referring energy dispersive X-ray spectra of (upper row) two typical soot like BB particles and (lower row) two typical K-rich BB particles from the investigated sample (Ni and Cu-peaks in the spectrum are due to the Ni-grids respectively the Cu-grid holder used in SEM analysis).

After these indications for potential BB influence within the tropopause region, we use the CARIBIC-AMS measurements of non-refractory compounds to estimate the amount of refractory aerosol (rA) which may correspond to black carbon (BC). Here, we reduce the background noise, especially of the organic signal, of the CARIBIC-AMS measurements by calculating mean values of the mass concentrations symmetrically over 3.5 min with a running mean (Box-Car method), i.e. over 7 individual data points. We averaged the UHSAS data over the same time interval and calculated the mass concentration from this mean UHSAS volume distribution, assuming spherical particles with a density of 1.5 g cm<sup>-3</sup>. The time series of the mass concentrations measured by CARIBIC-AMS and calculated from UHSAS data is shown in Fig. 7.

**Figure 7.** Time series of the GPS altitude (a), the total mass concentration of the AMS (b) and the UHSAS (b) and the individual mass concentrations for the AMS species (c). The AMS and UHSAS measurements are averaged over 3.5 min with the box-car method.

The estimation of the rA mass concentration is only possible, because both instruments measure in the same size range. We also have to emphasize that this estimation is only a best guess estimation with high uncertainties, based on several assumptions, and the estimated mass concentration has to be regarded as upper limit. The measurement uncertainties of the CARBIC-AMS are already in the range of 30 % (Canagaratna et al., 2007; Bahreini et al., 2009). The conversion of the number size distribution measured by the UHSAS to a volume distribution add a further considerable uncertainty. Since we used a difference between two measurements the calculated rA mass concentration can in principle include more than just one component, such as sea salt or mineral dust or BC. In the remote atmosphere, mineral dust and sea spray are typically found in the coarse mode above 1  $\mu$ m diameter, although some contribution to the submicron aerosol mass has also been observed (Brock et al., 2021). Furthermore, BB aerosol may also contain non-refractory salts such as KCl (Dang et al., 2022). The gained amount of rA is too high for real atmospheric values of BC in the order of less than 5 % (Yu et al., 2019). Therefore, this approximation can be regarded only with caution, but it yields also the potential for BC as well as the CO and the SEM particle analysis. The information on rA and potential black carbon may serve as an indicator of particle origin, especially regarding biomass burning. Furthermore, the estimated rA fraction is influenced by an instrumental drift of the CARIBIC-AMS in the beginning of the flight due to short preparation times with respect to reducing the background in the vacuum system.

Comparing both mass concentrations of the UHSAS and the CARIBIC-AMS (Fig. 7), we observe two different periods. Dur-

ing the first period, under tropospheric influence as evident from N<sub>2</sub>O and ozone and within the tropopause region (until 08:15 UTC), we found a difference between the two mass concentrations, indicating an instrumental drift but also a higher possible contribution of rA compared to the higher stratosphere. After 08:30 UTC, higher up in the stratosphere, we observe quite similar mass concentrations measured by the CARIBIC-AMS and the UHSAS and with this a much smaller possible contribution of the estimated rA. Compared to Fig. 5 we now analyze the complete flight with all altitude stacks. For this analysis, we divided the complete data set of F07 into five different regions, namely tropospheric background with N<sub>2</sub>O larger than 336 ppbv, UTLS with N<sub>2</sub>O between 330 and 336 ppbv, pollution events, stratospheric background with N<sub>2</sub>O between 330 and 325 ppbv (LMS) and stratospheric background with N<sub>2</sub>O lower than 325 ppbv (stratosphere) (see Fig. 8). This separation shows that within the UTLS including the polluted regions, we find higher estimated rA mass concentrations. The higher we get into the stratosphere, the less impact of polluted air masses is observed and we see an absolute increase of sulfate aerosol, which is typical for the stratosphere (Junge et al., 1961; Yue, 1981; Kremser et al., 2016).

Figure 8. Average of the absolute (a) and the relative (b) mass concentrations (including standard deviations shown as error bars) for the non-refractory aerosol species and the estimated amount of refractory aerosol (rA) during different measurement periods during F07. The separation of the regions is done by  $N_2O$  measurements and the exact times are provided in Tab. C2.

# 3.3 Air mass history and transport diagnostics

In the following, we present 10 day backward trajectories, initialized along the flight track, in combination with GFAS data provided by satellites, to link the observed pollution events and possible BB influence to potential source regions with active BB during the days before the measurement. The trajectories show that the polluted air masses during Flight 07 were located over Canada, mostly in the Canadian Arctic in the mid troposphere, 10 days before being sampled by TPEx F07. Over Central Canada, the air masses partly descend to lower tropospheric altitudes of around 3 km and cross regions with active forest fires ten to seven days before the TPEx measurements. Subsequently, the air parcels cross the North Atlantic towards Europe at altitudes below 5 km embedded in a potential dry air mass with respect to specific humidity (see Fig. 9 and Fig. F1). The compact trajectory bundle is uplifted on the edge of a WCB over Germany up to about 400 hPa within 12 hours and transported as a compact bundle towards the measurement location (see Fig. 10). In contrast to the pollution trajectories, those trajectories, which indicate pristine UTLS background, are also crossing Canada and regions with active wildfires, but in higher altitudes (Fig. 9b and Fig. 10). The tropospheric background trajectories however start at the east coast of Canada in low altitudes but in a region with mostly no fire activity. While crossing the Atlantic, the air masses can be found in distinct altitude regions. The trajectories with the biomass burning pollution are in the mid-troposphere and descend to 700 hPa on the way towards Europe. In contrast, the air masses measured in the UTLS background are 100 to 200 hPa higher in the free troposphere and the trajectories that end in the tropospheric regime are just below 700 hPa. In addition to differences of the altitude of the trajectories, we observed differences between upwind velocity during the uplift process into the UTLS. Here, the UTLS background and the polluted air masses are quite similar with a strong ascent over the Baltic Sea and Scandinavia whereas the tropospheric background trajectories are lifted at an earlier stage over the North Sea (Fig. 9 and Fig. 10).

**Figure 9.** Back trajectories for the different observed regimes ((a) pollution events, (b) UTLS background and (c) tropospheric background) in F07. The altitude of the trajectories is represented by the colorcode of the solid lines. Additionally, we added fire observation data provided by CWFIS (circles) in the vicinity of the trajectories. The size of the circles represents the size of the forest fire. Furthermore, we added the fire locations detected by the satellite in the GFAS retrievals of 10 June 2024 with fire radiative power (crosses).

**Figure 10.** Time series of pressure for the calculated LAGRANTO trajectories divided into the different regimes (Table C1): Polluted air masses (black), UTLS background (blue) and tropospheric background (red). The shaded area is the interquartile range (IQR) between the 25 and 75 % percentile with the median as solid line in the middle. The UTLS timeseries is shorter, because the trajectories leave the model domain 7 d before the measurements.

**Figure 11.** Top view of the first flight pattern within the restricted air space with deployed TPC-TOSS during F07. Colorcoded are measured quantities, such as the total aerosol particle number concentration by the UHSAS (a) and the vertical gradient of potential temperature (c). Furthermore, we analyzed meteorological parameters along the trajectories, namely the range between maximum and minium pressure (b) and the maximum specific humidity over the last 72 h before the measurements (d).

We use the meteorological data along the trajectories, to differentiate the air mass characteristics of the different air mass regimes. Therefore, we analyze the pressure difference between maximum and minimum pressure over the last 10 d as well as the maximum specific humidity over the WCB ascent period over the last 72 h before the measurements. Figure 11 shows that the observed pollution is found just in or directly next to a region where the air parcels experienced a strong vertical uplift of more than 400 hPa. This ascent ends one day before the measurements in the upper troposphere and allows time for subsequent small-scale cross-tropopause mixing. Regarding the moist processes within the WCB we are not able to identify significant differences between the polluted and unpolluted air masses (Fig. 11d), which leads to the conclusion that the wet deposition was in this case not efficient enough or other microphysics of BB aerosol is not resolved in the ICON model. As consequence we conclude that the BB aerosol can get transported to the UT and subsequently mixed into chemically stratospheric air masses. To summarize, the measurement data show that the observed polluted air masses in the stratosphere show elevated aerosol particle number concentration and CO mixing ratios. Furthermore, the measured particle size distributions show a size range typical for biomass burning aerosol as previously reported in the literature. Additionally, the differences between the mass concentrations measured by the CARIBIC-AMS and calculated from the UHSAS data indicate that refractory compounds,

potentially also BC, were measured in the tropopause region and especially during the pollution events. In addition to the insitu measurements, we analyzed filter samples for organic biomass burning tracers and impactor samples for individual particle composition to obtain more information on particles related to BB detected in the sampled air masses. Back trajectories started along the flight path show that the air masses originated from the lower troposphere over Canada, where active forest fires were reported and observed, and were lifted to the upper troposphere by a WCB over Europe about 1-2 days before the measurements. Although the process of WCB is typically accompanied by moist processes, the aerosol was transported into the UTLS and mixed into the chemical LMS. From previous studies (e.g., Ramanathan and Carmichael, 2008) it is known that the atmospheric lifetime of BC is typically less than 7 days, due to wet deposition processes like rainfall which occurs also in the process of WCB uplift. Nevertheless, we observe BB aerosol in the LMS where such particles have a radiative impact. In the following section we provide estimates for the radiative impact of this small scale aerosol streamer to assess the importance of WCB uplift processes for the radiative balance of the LMS.

## 3.4 Impact of potential BB pollution in the tropopause region

As pointed out by Ramanathan and Carmichael (2008) and Räisänen et al. (2022), BC has a positive radiative forcing at the TOA of up to 0.9  $\mathrm{Wm^2}$ . Ditas et al. (2018) show from regular observations on flights between Europe and North America that BB plumes show on average 0.14  $\mu\mathrm{g}\cdot\mathrm{m^{-3}}$  of BC and a typical UTLS background of 0.006  $\mu\mathrm{g}\cdot\mathrm{m^{-3}}$ . From these observations Ditas et al. (2018) derive an increase in the average heating rate of around 0.07  $\mathrm{K}\cdot\mathrm{d^{-1}}$  by the attribution of BC with a up to 0.44  $\mathrm{K}\cdot\mathrm{d^{-1}}$ .

In order to estimate the radiative effect of BB aerosol on the static stability in the present study, we use the available measurements of chemical composition and the dual platform measurements of temperature. The binned vertical profile of the relative mass concentration of the chemical composition including the potential for BC shows the highest estimated amounts around the chemical tropopause below 7500 m (see Fig. 12). Our upper limit estimation yields a maximum possible rA mass concentration of around 0.15  $\mu$ g (Fig. 8). As expected, with decreasing tropospheric influence the relative mass fraction of rA decreases with altitude as well and is close to zero above 9000 m. Including the large uncertainties of this upper limit estimation, our values are still in the range of BB-affected air, reported by Ditas et al. (2018) with 0.02  $\mu$ g and we can raise some hypothesis on the radiative impact on the tropopause region and its stratification.

**Figure 12.** Relative mass concentration for the non-refractory aerosol species (chloride in pink, ammonium in orange, nitrate in blue, organics in green and sulfate in red) and estimated highest possible refractory aerosol (rA) (black) in 500 m vertical bins between 7 and 10 km. The error bars represent the standard deviation in the altitude bins.

Therefore, we make use of the dual platform measurements of temperature and pressure to calculate vertical gradients in potential temperature  $(\theta)$ . We use these gradients to analyze the stratification of the tropopause region and derive a potential radiative forcing. Figure 11c shows the analyzed first pattern in the restricted air space colored with the derived vertical  $\theta$ gradient. In general, we observe the typical stable stratospheric stratification with positive vertical gradients of  $\theta$ . Besides the expected stable stratification, we observe a different stratification most prominent at the location of the polluted filament. Here, the gradient changes towards weaker stability close to  $0 \text{ K} \cdot \text{km}^{-1}$ . The location of this anomaly is the same as for the polluted air masses with BB influence (Fig. 11c). This leads to the assumption that this change is related to the streamer of polluted air masses. The observed change in the vertical  $\theta$ -gradient reaches up to 2 K · km<sup>-1</sup>. From the trajectories we find that after the WCB uplift there is no influence of cloud liquid and ice water in the UTLS for the last 18 h before the measurements. Additionally, the trajectories indicate a quasi-isentropic transport in the UTLS from the uplift area to the measurement locations with a slight increase in the static stability and PV. However, it is important to say that the trajectories do not include any aerosol data, e.g. aerosol radiative heating. Therefore, considering these observations in the trajectory data started at the Learjet position, we hypothesize that the changes in the static stability may also be forced by radiative effects of the transported rA in the absence of (cirrus) clouds. Referring to the observations by Ditas et al. (2018) typical heating rates as consequence of BC are in the magnitude of  $0.07~{\rm K}\cdot{\rm d}^{-1}$  up to  $0.44~{\rm K}\cdot{\rm d}^{-1}$  in extreme cases, which shows a significant contribution to the radiative feedback on lower stratospheric dynamics. Combining the uncertainties of the rA approximation with potentially included BC and the gradient calculation of potential temperature (31 %, see Bozem et al. (2025)) we end up in the same regime of expected heating rates of roughly  $0.1 \text{ K} \cdot d^{-1}$  of the tropopause region after the WCB uplift.

As the transport of BC and BB plumes was mostly studied in the presence of pyroconvection and fast uplift of pollutants to the tropopause region, current studies like this and that of Khaykin et al. (2025) show an additional transport pathway of BB aerosol into the UTLS and especially the LS. This uplift can occur close to the fire locations like in Khaykin et al. (2025) or after low-level long-range transport far away from the pollution source as shown in this study.

#### 4 Conclusions

The influence of WCBs on the UTLS aerosol composition is only rarely discussed in previous studies (e.g., Voigt et al., 2017; Trickl et al., 2024). In our study, we present a case-study of aerosol characteristics in a WCB outflow region over Germany. 415 Here, we observed a small-scale streamer of polluted air masses, more precisely elevated aerosol number concentration between 100 nm and 1 um with CO mixing ratios larger than 100 ppby in the chemically stratospheric air (N<sub>2</sub>O < 336 ppby). Further, analysis of the particle size distribution in the pollution event shows a maximum of particles with a diameter between 200 and 400 nm, which is a characteristic of biomass burning size distribution. Further evidence for the influence of biomass burning is found by the offline analysis of impactor samples using a SEM. Here, we find particles which refer to biomass burning, such as soot and carbon dominated particles with larger fractions of potassium. LAGRANTO back trajectories on the 420 basis of ICON global analysis wind fields for the last 10 days before measurement also show biomass burning over Canada as potential source for the observed pollution. However, these trajectories are not able to resolve convective transport either in pyro-convection, boundary-layer convective clouds, or embedded convection in the WCB. According to the trajectory data, the air masses observed by TPEx F07 originate in the lower troposphere over Canada and the Canadian Arctic and crossed spots of active forest fires. These polluted air masses were then transported across the Atlantic within the lower troposphere before 425 subsequent uplift embedded on the edge of a WCB over Europe. This uplift was strong enough to transport the BB aerosol into the UTLS, and after end of ascent likely trigger stratosphere-troposphere exchange. Additionally, we analyzed the chemical composition of the aerosol in the pollution event and during background conditions in the troposphere, the undisturbed LMS and the stratosphere. Here, we estimated the mass concentration of rA by the difference between the total mass measured by the AMS and the UHSAS, under the assumption of a density of 1.5 g cm<sup>-3</sup>. This method yields the highest amounts of rA 430 around the tropopause and rather no rA in the stratosphere. Besides the in-situ measurements of the chemical composition of aerosol particles and trace gases, we used offline filter analyses of collected aerosol particles during the flight. The analysis of these filters needs to be considered with some caution, as consequence of a high background signal for these two flights. But even with this caution, we are able to see some indications for the influence of BB on this day, which supports the found results 435 for BB as source of the pollution.

In addition to the chemical composition and aerosol measurements, we take advantage of the dual platform measurements of temperature and pressure to derive the vertical gradient in potential temperature to describe the radiative impact of this BB intrusion on the UTLS. From this, we observe a change in the gradient of  $\theta$  by around  $2 \text{ K} \cdot \text{km}^{-1}$ , with a contribution which is caused most likely by the observed rA and BC in this region.

40 Finally, we were able to show an additional pathway of BB pollution into the extratropical tropopause region by WCB uplift.

In contrast to other studies (Khaykin et al., 2025), we observe this uplift mechanism after low-level long-range transport. The radiative feedback of such transported rBC into the tropopause region now needs to be calculated to show the impact of this additional pathway.

**Figure 13.** Schematic overview of the observed processes in our study. The polluted air masses from Canadian forest fires travels in the lower and free troposphere across the Atlantic ocean, before strong ascent associated with a WCB over Europe occurs. As consequence of this strong uplift to the tropopause region the polluted air masses are in a region of stratosphere-troposphere exchange and are mixed with chemically stratospheric air masses.

Data availability. The measurement data, model data and trajectory data along the flight path is published on Zenodo (Lachnitt, 2025; Miltenberger, 2025).

Author contributions. PJ set up the study, analyzed the data and wrote the manuscript. PJ, JS, JW are responsible for the AMS and UHSAS data. NE, IK, HB and PH provided CO, N<sub>2</sub>O and O<sub>3</sub> data. DK and HCL provided ECMWF data along the flight path. YL provided temperature and humidity data. CR and NS provided the H<sub>2</sub>O data. AM and CS provided ICON-EU WCB diagnostics and the LAGRANTO trajectory data. AB and AV sampled and analyzed the SOAP samples. SR provided the mCPC data. ME, SI, KK and LS are responsible for the impactor samples and the analysis in the SEM. All authors contributed and commented on the manuscript.

Competing interests. Some authors are members of the editorial board of journal Atmospheric Chemistry and Physics.

Acknowledgements. The authors acknowledge the team of enviscope GmbH and GFD GmbH for the opportunity to carry out the campaign and the technical support during the campaign. Additionally, the authors thank the graphical office of the MPI-C for the schematic drawing. Furthermore, the authors appreciate the support by Tobias Könemann and Jesse Steiner (both DMT) for their technical support with the

455

PJ, JS, SB, AB, AV, CS, AM, HCL, DK, LS, SI, KK, ME, NE, PH, SR and JC acknowledge funding by the Deutsche Forschungsgemeinschaft (DFG, German Research Foundation) – TRR 301 – Project-ID 428312742: "The tropopause region in a changing atmosphere"

UHSAS systems. This publication is generated using Copernicus Atmosphere Monitoring Service Information [2024].

## 460 Appendix A: Identification of the chemical stratosphere

For this study, we use ozone  $(O_3)$  and nitrous oxide  $(N_2O)$  to determine the chemical tropopause as reference for tropospheric and stratospheric air masses. This chemical tropopause, based on the mixing ratios and the vertical gradient of both trace gases, is around 308 K potential temperature.

Figure A1. Vertical profiles of  $O_3$  (a) and  $N_2O$  (b) for the complete F07, colorcoded with the total aerosol number concentration measured by the UHSAS. The dashed lines visualize the chemical tropopause based on the vertical gradient and the mixing ratios of the individual species.

#### **Appendix B: Identification of mixed air masses**

In Appendix B, we introduce tracer-tracer correlation to identify mixing processes between the troposphere and the stratosphere. Therefore, we use the CO versus N<sub>2</sub>O (Fig. B1a) correlation. This correlation has also the advantage, that we are able to determine the tropospheric end member of each individual mixing line, which occurs in the correlation. As colorcode we use the information about the flight pattern within the restricted air space, to better identify small-scale mixing occurrence. We observe the most dominant perturbation of the overall mixing line during the first pattern, where the correlation describes a "z-shape" structure. The tropospheric CO end member for this part shows CO mixing ratios over 150 ppbv which is a distinct

marker for mixed pollution into the LMS.

In the correlation between total particle number concentration N<sub>UHSAS</sub> versus N<sub>2</sub>O (Fig. B1b), we identify a mirrored "C-shape" over the complete flight, which is also expected from previous studies in literature (e.g., Borrmann et al., 1993). Here, we observe a similar perturbation during the first pattern with particle number concentration exceeding the background values of the correlation and reaching up to tropospheric values.

Figure B1. Tracer-Tracer correlation of CO vs.  $N_2O$  (a) and N vs.  $N_2O$  (b). The colorcode indicates the individual flight pattern in the restricted air space, each pattern is a separate round. The dashed line marks the chemical tropopause.

## Appendix C: Definition of the analysis regions and averaging intervals

This section gives detailed information about the time intervals used for averaging the size distributions in Fig. 5 as well as the regions from Fig. 8. We chose 10 s, in order to represent only the main total number concentration peak of the polluted air masses for the size distributions. For comparison between the different measurement regions and events we used the 10 s for all intervals, even if enlarged periods for the background periods had been possible.

The following time periods are used for the averaged size distributions:

480

| Event/Region            | Times (UTC)       |
|-------------------------|-------------------|
| Pollution               | 07:25:27-07:25:37 |
| UTLS background         | 07:26:05-07:26:15 |
| Pollution               | 07:27:49-07:27:59 |
| tropospheric background | 07:32:48-07:32:58 |
| Pollution               | 07:43:31-07:43:41 |
| UTLS background         | 07:44:12-07:44:22 |
| Pollution               | 07:46:32-07:46:42 |

**Table C1.** Time periods for the averaged size distributions shown in Fig. 5.

As a consequence of the 30 s time resolution of the AMS, there is no equal distribution of data points between the different measurement regions. The scale of the pollution streamers was too small for obtaining more than three AMS data points for each pollution event. The following time periods are used in the specific region definitions:

| Region                               | Times (UTC)         |
|--------------------------------------|---------------------|
|                                      | 07:20 - 07:23:30    |
| Troposphere                          | 07:30:30 - 07:40:30 |
| $N_2O > 336 \text{ ppbv}$            | 07:49:00 - 08:00:00 |
|                                      | 09:18:30 - 09:30:00 |
| UTLS $330 < N_2O < 336 \text{ ppbv}$ | 07:24:00 - 07:25:00 |
|                                      | 07:26:30 - 07:27:30 |
|                                      | 07:29:00 - 07:30:00 |
|                                      | 07:41:00 - 07:42:30 |
|                                      | 07:44:30 - 07:46:00 |
|                                      | 07:47:30 - 07:48:30 |
|                                      | 08:00:00 - 08:05:00 |
|                                      | 08:16:30 - 08:21:00 |
|                                      | 09:08:30 - 09:18:00 |
| Biomass Burning filament             | 07:25:30 - 07:26:00 |
|                                      | 07:28:00 - 07:28:30 |
|                                      | 07:43:00 - 07:44:00 |
|                                      | 07:46:30 - 07:47:00 |
|                                      | 08:05:30 - 08:16:00 |
| LMS                                  | 08:23:00 - 08:36:00 |
| 325 < N <sub>2</sub> O < 330 ppbv    |                     |
| Stratosphere                         | 08:37:30 - 09:05:00 |
| $325 > N_2O$                         |                     |

**Table C2.** Time periods for the averaged chemical composition measurements shown in Fig. 8.

## Appendix D: Offline filter measurements

To obtain additional information on the organic particle composition, we use filter samples from the SOAP instrument. Note that the filter samples can not be allocated to certain positions of the aircraft, because of the long sampling periods of more than one hour for each filter. These extended periods were necessary to collect sufficient material on the filters, ensuring an adequate signal-to-noise ratio for the subsequent analysis. For this reason, small-scale pollution events are difficult to cover and challenging to detect. Not only the small-scale events but also the dilution within the atmosphere can lead to very low concentrations of certain compounds. Nevertheless, the sampled filters provide some indications of whether the air masses might be influenced by BB or not. For this purpose an analysis using high-resolution mass spectrometry for a selection of BB tracers was performed on the filter extracts. In our analysis these BB tracers are levoglucosan, 4-nitrophenol, phthalic acid and vanillic acid, which are already known as common tracers from previous studies (e.g., Simoneit and Elias, 2001; Bluvshtein et al., 2017; Wang et al., 2022). The sampling times of the filters are summarized in Tab.D1. For the analysis of the filter samples we also need to take Flight 08 into account, because the reference blank filter was also flown during F08. This flight took place directly after Flight 07 and probed partly the same air masses of the earlier flight over the North Sea and Western Germany. Still, the flight blank should provide information about any contamination that might have happened. Some more information on Flight 08 are given in appendix E.

| Filter        | Times (UTC)                  |
|---------------|------------------------------|
| F07, Filter 1 | 07:14 - 08:05; 09:10 - 09:31 |
| F07, Filter 2 | 08:08 - 09:03                |
| F08, Filter 1 | 11:25 - 12:58                |
| F08, Filter 2 | 12:58 - 14:38                |

**Table D1.** Sampling times of SOAP filters during the double flight F07 and F08 on 17 June 2024. F07, Filter 1 corresponds to the time series in Fig. 3.

**Figure D1.** Results of filter analysis for BB species on the probed samples during the double flight on 17 June 2024. The signal of the flight blank was subtracted and the measurements were normalized according to the sampling volume (a). (b) the mean mass concentration, including the standard deviation, of the CARIBIC-AMS data and the estimated amount of rA for the sampling periods of the filter substrates.

A qualitative analysis of the four biomass burning tracers indicated that only two of the tracers were detected on the filter samples (Fig D1). The strongest signal occurs for phthalic acid during Flight 08 followed by a clear signal of vanillic acid during Flight 07. However, the detection of phthalic acid and vanillic acid does not coincide with the observed transient BB periods. Likely these events are too short to become visible by offline analysis. All data are blank corrected with the respective flight blank. A small signal of levoglucosan appears on three out of four samples. However, the levoglucosan results are uncertain due to high background concentrations on the blank filter. The long atmospheric transport of the BB plumes and short lifetimes of the organic species, especially for levoglucosan (0.5 up to 4 days (Hoffmann et al., 2009)), might also explain the low abundance of levoglucosan during the described events. Contamination of the filter substrates due to the double flight without the possibility to exchange the filters in between also resulted in significant blank signals that were subtracted from the samples. The flight blank shows significantly higher intensities than the solvent blank, which indicates that contamination might have occurred and improving sampling strategies is essential. Nevertheless, we can say with caution for the given reasons that the air masses which were probed on the 17 June 2024 show influence of BB to some extent, but there are differences in the processing of the air masses during the transport towards Europe.

# 515 Appendix E: Further information on F08

In this section of the appendix, we show some additional data from F08 as motivation to compare it with the presented flight F07. The flight pattern was from the North Sea towards Southwest Germany, close to Koblenz, along the tropopause and in the stratosphere to probe enhanced mixing. In the region over the North Sea the air masses are similar to the probed air masses in the morning flight.

**Figure E1.** Time series of the in-situ measurements during F08 on 17 June 2024. The top panel shows the GPS altitude, the second panel shows in-situ trace gas measurements of CO (black),  $N_2O$  (green) and water vapor (blue). The third row shows the measured mass concentration by the AMS (red) and the UHSAS with a density estimation of 1.5 (black). The lowest panel displays the individual mass concentration from the AMS.

Figure E2. Flightpath of F07 (dashed) and F08 (solid) colorcoded with the GPS altitude.

## 520 Appendix F: Additional trajectory information

Here, we show the air mass history of the LAGRANTO trajectories for the single days before the measurement in the large frame with CAMS forecast data with a 0 h lead time for 00:00 UTC each day.

**Figure F1.** Combination of CAMS forecast data (colored background) and daily split trajectory data (yellow lines). All panels show the CAMS forecast with 0 h lead time for specific humidity (q) on 850 hPa (a-i) and 400 hPa (j-k). The trajectories are split for each 24 h interval to identify the large-scale context of the trajectories.

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
