# Peer review of "Transport of Biomass Burning Aerosol into the Extratropical Tropopause Region over Europe via Warm Conveyor Belt Uplift"

_EGUsphere, 2025_

## Referee Comment (RC1)

**Review of the ACP manuscript acp-2025-1346**

"Transport of Biomass Burning Aerosol into the Extratropical Tropopause Region over Europe via Warm Conveyor Belt Uplift"

By Philipp Joppe, Johannes Schneider, Jonas Wilsch, Heiko Bozem, Anna Breuninger, Joachim Curtius, Martin Ebert, Nicolas Emig, Peter Hoor, Sadath Ismayil, Konrad Kandler, Daniel Kunkel, Isabel Kurth, Hans-Christoph Lachnitt, Yun Li, Annette Miltenberger, Sarah Richter, Christian Rolf, Lisa Schneider, Cornelis Schwenk, Nicole Spelten, Alexander L. Vogel, Yafang Cheng, and Stephan Borrmann.

The above manuscript describes aircraft-borne aerosol particle measurements conducted in the upper troposphere and lowermost stratosphere over Europa. The analysis focuses on a short time period of one of the TPEx campaign flights, when the outflow region of a warm conveyor belt (WCB) was probed and influence from biomass burning was found. Overall, the manuscript is well written, the set-up with towed sensor shuttle is relative unique, and the supporting meteorological data are very useful. In contrast to previous studies, the detected biomass signature stems from smaller wildfire events which reach the tropopause region due to the WCB and not by their own dynamics (i.e. as pyroconvection). As this is somehow new and could be relevant to the UT/LS chemistry or UT/LS influence on the radiation budget, the manuscript should be published. However, there are a few questions concerning the data processing and the interpretation of the results. Moreover, the prove of the relevance of such kind of events is missing, which diminishes the value of the results (are they relevant or not?).

**Specific remarks:**

p. 1, l. 9: I can see the "800 particles per $cm^3$" in Fig. 3, but for me the background is about 200 particles per $cm^3$, hence at least a factor of four lower, not two.

p. 2, l. 24: the sulfate aerosol and the BC you refer to, which one is it? The average total atmospheric amount? Please specify.

p. 2, l. 35: Please cite not only references from your group, there are other publications which have shown which are related to your topic, for instance Brioude et al, Atmos. Chem. Phys., 7, 4229–4235, 2007 or Zahn et al., J. Geophys. Res. 105, 1527-1535, 2000.

p. 3, l. 57: The sentence on the BC lifetime, this hold true for other particles as well. And after modification, please move it somewhere else (or delete it), as it does not fit to the rest of the paragraph.

p. 3, l. 74: You focus on a dedicated, short section of one flight. But looking at the flight pattern and having in mind that there was a second flight on the same day, did you check your data for the chance of having probed the airmass a second time, later (or earlier) in the campaign? On page 15 you suggest that partly probing the same airmass again have been the case.

p. 3, l. 78: The statement of the "highly variable tropopause altitude", well, it looks variable, but not highly variable, e.g. there is no tropopause fold. Please modify the statement or justify why it is valid.

p. 5, l. 97: Even if the particle size range which is most interesting for you is less prone to particle losses in your inlet system and sampling line, you must at least provide an estimate or an upper limit on the respective particle losses.

p. 5, l. 105: Related to the point above, how about an in-flight intercomparison between the TOSS and the Learjet instruments? There must be a flight condition, where this should have been possible and this would highly increase confidence in the data quality. Similar to the radiation calculations later on, it is not sufficient to refer to a potential future paper.

p. 6, l. 108: Again, the statement on the upper inlet cut, which analysis is it based on?

p. 6, l. 115: You state that you operated 3 CPCs, but you provide only two lower threshold diameters. Why?

p. 6, l. 120: If I´m not totally wrong, 5 km in 30 s would result in a TAS of 167 m/s, which seems to be very slow for a jet aircraft in the upper troposphere. Are you sure the number is correct?

p. 7, l. 154: I did not find the time resolution of the filter sampling, please provide this information.

p. 7, last paragraph: I´m not an expert in this, but can you exclude particle changes during the storage times? Either way, could you please write a sentence, if these can be excluded (and if yes, why) or if they are of minor importance or …

p. 9, l. 214: Same statement as in the first comment, I do not see this factor two, it is at least a factor of four.

p. 9, l. 225: If I´m not totally wrong a $H_2O$ mixing ratio of 100 ppmv is rather typical for the midlatitude summertime UT and not an indicator for stratospheric air, or?

p. 11, l. 250: Why do you provide the information that the flight pattern was flown after the TOSS deployment? Do you want to say, that the TOSS was not applied during that flight pattern? If so, please state it in that way. Same in the figure caption of Fig. 4.

p. 11, Fig. 5.: This figure puzzles me a lot. First of all, why is the first a delay in the AMS data and in the next peak the AMS is ahead? Secondly the strange looking UT/LS background volume size distribution, why are there jumps of 50% in relative narrow size bins? Is there an issue with the assumed refractive index of the particles? And the error bars are misleading, the measurement period is short, hence it would be much better to indicate the measurement uncertainty here, which should be some ten percent, I guess. Volume size distributions derived from OPC measurements are highly uncertain! The data behind this figure need a deeper analysis. The different colors in Fig. 5b are not explained in the legend.

p. 12, last paragraph: You argue here and in the following that you have at least up to the LMS a non-negligible amount of soot in your particles (see also Fig. 7). And in the troposphere it seems to be (for me) unrealistically high. This will definitely affect your optical particle measurements, was this considered in your data processing? Otherwise you cannot trust the distributions in Fig. 5b.

p. 16, Fig. 8: I understand the "normalized by sample volume" on the y-axis (please remove the empty parenthesis), but I do not understand the "normalized to the flight blank and sampling time" in the figure caption. And these are two different statements, right?

p. 16, l. 319: The interpretation of the particulate BB tracers, what do the results indicate? That there have always been BB influence during the sampling period?

p. 16, l. 323: The short lifetime of levoglucosan is how long? Please provide this number in the text.

p. 19, l. 374: The whole paragraph. Either you show results of the radiative impact here or you give an estimate on how frequent such events might occur. Otherwise the value of your results is hard to estimate, i.e. are they relevant.

p. 20, Fig. 11: Again I doubt the BC fraction of 40% at the lowest flight level. I assume that there must have been other particle material like dust.

p. 28, Fig. D1: The first too low AMS mass concentrations, could these be caused by the AMS warm-up time?

**Technical corrections:**

p. 3, l. 76: Please change to "The TOSS and the aircraft were equipped …"

p. 3, l. 81: Please specify what is meant with "stratospheric intrusions which increase during the flight", are they reaching deeper into the troposphere or are they covering a larger area or do they occur more frequently?

p. 4, Fig. 1: The flight path is displayed in red, not in black, as stated in the figure caption.

p. 6, l. 132: Please provide a reference publication for the instrument and the uncertainties.

p. 7, l. 149: The first two sentences of this paragraph provide partly the same information. Please remove this doubling.

p. 8, l. 208: The information that you use $N_2O$ for defining the chemical tropopause is already given in line 203. Please remove one of the two sentences.

p. 9, l. 211: This subsection is quite long, do you see any chance to split it? This would make it easier for the reader to follow your line of arguments.

p. 9, l. 235: The "recent particle formation event" could it be an aircraft plume encounter? Did you check for instance flightradar24 for such a possibility?

p. 10, l. 237: "the chemical stratosphere" is probably not the correct term (what would this be?), you mean "chemically stratospheric air".

p. 11, Fig. 4: Please provide the particle size range information in the legend of Fig. 4a.

p. 13, l. 285: You refer to Fig. D1 and discuss it here in the main text. Consequently, the figure should be shown here. Same for figure F2 later on.

p. 17, l. 346: The half sentence "and the biomass burning pollution trajectories" does not fit here and seems to me remnant from a former test version.

p. 18, l. 359: Please change "aerosol number concentration"
to "aerosol particle number concentrations".

p. 21, l. 407: "the chemical … stratosphere" is wrong again, you mean "in chemically stratospheric air".

p. 21, l. 429: Please exchange "into" with "on", otherwise the sentence does not make sense.

p. 28, l. 477: there is a space missing in-between "gcm" and also in ".5min" in the next line.

p. 31, Fig. F1: the red line indicating the back trajectories are hard to see over the orange/brown background. Please use a different color, e.g. bright green.

---

## Referee Comment (RC2)

**Anonymous Review of *Transport of Biomass Burning Aerosol into the Extratropical Tropopause Region over Europe via Warm Conveyor Belt Uplift**

Anonymous Reviewer

June 26, 2025

**1 Summary**

In this work, Joppe *et al.* studied the transport of biomass burning aerosol into the extratropical upper troposphere and lower stratosphere (exUTLS) through a warm conveyor belt. This was a case study of one event that occurred during Flight 7 in the TropoPause compositon gradients and mixing Experiment (TPex) campaign.

During the TPex campaign, a Learjet 35A flew a package of in-situ and offline measurements. The in-situ aerosol measurements were sizing from a UHSAS, OPC, and mcCPC, as well as non-refractory aerosol chemical composition from a miniAMS. The in-situ gas phase measurements were $N_2O$, CO, and $O_3$. There were also offline aerosol measurements. These include a filter-based collection that was rinsed and run through an HPLC and then through an orbitrap. There was also a cascade impactor that held TEM grids for offline SEM-EDX analysis. Finally, a similar suite of in-situ measurements was taken on the Towed Sensored Shuttle (TOSS) to measure vertical gradients.

Overall, this is a very well-written paper with very few technical comments. However, I have two major comments that, if addressed, I think will increase the clarity of the paper and strengthen its message.

**2 Major Comments**

1. My first major comment is on the rBC calculation. Refractory BC is generally a small component ($<5\%$) of fresh biomass burning smoke. For example, the study by Yu et al. shows that freshly injected pyroCb smoke in the stratosphere from the Pacific Northwest Event was $\sim2\%$ rBC, which accounted for its incredible rise to 23 km. The estimate that rBC may be

almost 40% and 20% of the tropospheric and UTLS aerosol mass, respectively, is difficult to reconcile with many of the previous in situ measurements at these altitudes (see ATom and HIPPO campaigns). The authors are careful to point out their rBC calculation is an upper estimate, but I think these estimates are so far off that this warrants further explanation.

2. The above major point affects the heating rate calculated in Section 3.3, and will also affect the radiation simulations cited at the end of Section 3.3.

3. The other major comment is that the SOAP analysis, in which the filters were washed and analyzed with HPLC and orbitrap mass spectrometry, seems relatively weak to me. The CO + UHSAS, AMS, TEM grid and back trajectory analysis are all much stronger evidence that this pollution event is from biomass burning. Certainly, the technique seems strong, but I feel that removing these results from the main paper would tighten its focus and make it stronger. This reviewer suggests that they can be moved to the supplemental section if needed.

**3 Minor/Technical Comments**

- P11L244: In Figure 4, it is unclear to me whether the data in all the plots is from the TOSS or from the Learjet cabin.

- P12L272: How many of the grid particles were non-refractory vs refractory?

- P12L274: Are soot particles considered refractory or non-refractory in this analysis? More details regarding the TEM grid analysis needs to be outlined either here or in the experimental section.

- P13L284: There needs to be a space between g and $cm^{-3}$.

- P14L291: If you look at Figure 11 of Brock 2021, you can see that there is some contribution from sea salt and dust to the submicron aerosol mass. Furthermore, there are likely to be non-refractory salts like KCl in biomass burning aerosol.

---

## Author Response (AR1)

**ACP manuscript acp-2025-1346**

Transport of Biomass Burning Aerosol into the Extratropical Tropopause Region over Europe via Warm Conveyor Belt Uplift

P. Joppe et al.

**Author comments to Reviewer #3**

The reviewer comments are written in this font style and color.

Our answers are written in this font style and color.

Changes in the revised version of the manuscript are written in red.

The above manuscript describes aircraft-borne aerosol particle measurements conducted in the upper troposphere and lowermost stratosphere over Europa. The analysis focuses on a short time period of one of the TPEx campaign flights, when the outflow region of a warm conveyor belt (WCB) was probed and influence from biomass burning was found. Overall, the manuscript is well written, the set-up with towed sensor shuttle is relative unique, and the supporting meteorological data are very useful. In contrast to previous studies, the detected biomass signature stems from smaller wildfire events which reach the tropopause region due to the WCB and not by their own dynamics (i.e. as pyroconvection). As this is somehow new and could be relevant to the UT/LS chemistry or UT/LS influence on the radiation budget, the manuscript should be published. However, there are a few questions concerning the data processing and the interpretation of the results. Moreover, the prove of the relevance of such kind of events is missing, which diminishes the value of the results (are they relevant or not?)

Thank you very much for the detailed reading of the manuscript and the positive consideration of the topic for publication. Thanks also for the detailed questions and comments which help to improve the manuscript.

**Specific remarks:**

p. 1, l. 9: I can see the "800 particles per cm3" in Fig. 3, but for me the background is about 200 particles per cm3, hence at least a factor of four lower, not two.

We checked the exact values in the time series. The pollution shows an enhancement of 2.5 compared to the UTLS background and up to 4 compared to the tropospheric background. For clarification we changed the phrasing to:

This is higher by a factor of more than two compared to the UTLS background and up to a factor of 4 higher than the tropospheric background.

p. 2, l. 24: the sulfate aerosol and the BC you refer to, which one is it? The average total atmospheric amount? Please specify.

Thank you for the careful reading. Yes, we refer to the average total atmospheric amount. We added this information to the revised version of the manuscript.

For example, at the top of the atmosphere (TOA), the global radiative effect of sulfate aerosol is a cooling effect up to about -1.3 Wm-2 whereas the global radiative effect of black carbon (BC) shows a strong heating effect of up to 0.9 Wm-2 (Masson-Delmotte et al., 2023; Kalisoras et al., 2024; Ramanathan and Carmichael, 2008; Räisänen et al., 2022).

p. 2, l. 35: Please cite not only references from your group, there are other publications which have shown which are related to your topic, for instance Brioude et al, Atmos. Chem. Phys., 7, 4229–4235, 2007 or Zahn et al., J. Geophys. Res. 105, 1527-1535, 2000.

We agree with the reviewer that the references should be neutral and not only from the own group. Thank you to the reviewer for suggesting us these publications, which were new for us. We added these references.

p. 3, l. 57: The sentence on the BC lifetime, this hold true for other particles as well. And after modification, please move it somewhere else (or delete it), as it does not fit to the rest of the paragraph.

We deleted this sentence in order to stay within the focus of this paragraph, which deals with OA

p. 3, l. 74: You focus on a dedicated, short section of one flight. But looking at the flight pattern and having in mind that there was a second flight on the same day, did you check your data for the chance of having probed the airmass a second time, later (or earlier) in the campaign? On page 15 you suggest that partly probing the same airmass again have been the case.

Unfortunately, we were not able to fly the days before F07 to probe the air mass during uplift and WCB transport, due to airport closure over the weekend. On the following day, we conducted a flight, but with focus on PBL characterization and convective uplift over central Germany. Thus, this flight probed completely different air masses. Therefore, the only possibility of probing partly the same air mass is the already written suggestion.

p. 3, l. 78: The statement of the "highly variable tropopause altitude", well, it looks variable, but not highly variable, e.g. there is no tropopause fold. Please modify the statement or justify why it is valid.

We changed the statement to only a variable tropopause in the measurement region.

The goal of research flight F07 was to probe a region with a variable tropopause altitude (see Fig. 1a). As consequence, we expected enhanced cross-tropopause mixing as a consequence of a

low-pressure system over the North Sea west of Norway and predicted low Richardson numbers in the restricted air space (not shown).

p. 5, l. 97: Even if the particle size range which is most interesting for you is less prone to particle losses in your inlet system and sampling line, you must at least provide an estimate or an upper limit on the respective particle losses.

We added a corresponding sentence regarding the particle losses at the end of the inlet description.

In the measured size range of the UHSAS we calculated transmission efficiencies of 86 % at the boundaries and 95 % at diameters around 300 nm. These calculations were performed for an ambient pressure of 300 hPa and 240 K using the Particle Loss Calculator by von der Weiden et al. (2009).

p. 5, l. 105: Related to the point above, how about an in-flight intercomparison between the TOSS and the Learjet instruments? There must be a flight condition, where this should have been possible and this would highly increase confidence in the data quality. Similar to the radiation calculations later on, it is not sufficient to refer to a potential future paper.

We have an in-flight intercomparison between the TOSS and the Learjet instruments. This intercomparison as well as the complete characterization of instruments are described in Bozem et al. (2025) which is now available as preprint. Therefore, we updated the reference to the preprint.

p. 6, l. 108: Again, the statement on the upper inlet cut, which analysis is it based on?

We performed particle loss calculations by the Particle Loss Calculator by von der Weiden et al., 2009. For this, we transferred the inlet metrics into the software for calculation. For clarification we added the following sentence:

The particle loss calculation was done using the Particle Loss Calculator described in von der Weiden et al., 2009.

p. 6, l. 115: You state that you operated 3 CPCs, but you provide only two lower threshold diameters. Why?

To cross-check the data quality of the aerosol number concentration during the flights, two of the three mc-CPC channels were operated at the same dT and thus at the same cutoffs. For reasons of clarity, however, we only show one channel here.

p. 6, l. 120: If I'm not totally wrong, 5 km in 30 s would result in a TAS of 167 m/s, which seems to be very slow for a jet aircraft in the upper troposphere. Are you sure the number is correct?

Yes, this is right. The TAS is lower than for typical jet aircraft in the UTLS. This comparatively low air speed is forced by technical constraints when the TOSS is deployed and towed below the Learjet. The reported TAS is taken from the Learjet avionic data system.

p. 7, l. 154: I did not find the time resolution of the filter sampling, please provide this information.

The filter sampling has no fixed time resolution. The sampling period depends on the meteorological and expected conditions, such as troposphere, tropopause region, stratosphere or expected polluted regions from the CAMS forecast with adjustments based on the in-situ measurements. As we moved the SOAP analysis to the appendix, referring to Reviewer#4 a detailed overview over the filter sampling periods is provided in Table D1.

p. 7, last paragraph: I'm not an expert in this, but can you exclude particle changes during the storage times? Either way, could you please write a sentence, if these can be excluded (and if yes, why) or if they are of minor importance or ...

Thank you for raising your concerns about artifacts that might occur due to the storage. It indeed can happen. However, here we want to refer to Resch et al. 2023 (<a href="https://doi.org/10.5194/acp-23-9161-2023">https://doi.org/10.5194/acp-23-9161-2023</a>,), who did a thorough study on storage conditions. Based on this we stored our filters below 0 °C in order to prevent reactions occurring on the filter or losses. More details can also be found by the added reference of Breuninger et al., 2025 which describes the filter sampling in more detail.

p. 9, l. 214: Same statement as in the first comment, I do not see this factor two, it is at least a factor of four.

We reformulated the sentence to the following, also referring to the reply to the first comment.

[...] because the time series shows a very small-scale pollution event with an increase in particle number concentration by more than a factor of two compared to the UTLS background.

p. 9, l. 225: If I'm not totally wrong a H2O mixing ratio of 100 ppmv is rather typical for the midlatitude summertime UT and not an indicator for stratospheric air, or?

Yes, this is correct and 100 ppmv is more upper tropospheric than lower stratospheric. We rephrased the paragraph to show the mixing of chemically stratospheric air (N2O and O3) with UT air masses (H2O).

The interpretation of the mixed air masses into chemically stratospheric air is also supported by the O3 mixing ratios above 150 ppbv and H2O values near 100 ppmv H2O.

p. 11, l. 250: Why do you provide the information that the flight pattern was flown after the TOSS deployment? Do you want to say, that the TOSS was not applied during that flight pattern? If so, please state it in that way. Same in the figure caption of Fig. 4.

This information is provided, because the TOSS can only be detached from the aircraft and operated in the restricted air space. Further, we want to express by this sentence that we analyzed the part of the flight during which the TOSS was already running and measuring. The additional information from the TOSS is needed to calculate vertical gradients of potential temperature over the whole pattern shown in Fig. 4.

p. 11, Fig. 5.: This figure puzzles me a lot. First of all, why is the first a delay in the AMS data and in the next peak the AMS is ahead? Secondly the strange looking UT/LS background volume size distribution, why are there jumps of 50% in relative narrow size bins? Is there an issue with the assumed refractive index of the particles? And the error bars are misleading, the measurement period is short, hence it would be much better to indicate the measurement uncertainty here, which should be some ten percent, I guess. Volume size distributions derived from OPC measurements are highly uncertain! The data behind this figure need a deeper analysis. The different colors in Fig. 5b are not explained in the legend.

Thank you for the careful reading of the figures in the manuscript. We hope we can deliver some helpful explanations to this figure in the following:

The delay in the AMS data or the shift of the data has two reasons which partly overlap. First, we averaged the AMS signal over 3.5 minutes with a running mean to reduce the noise and obtain signals above the detection limits. This is the largest factor for the shift.

Regarding the size distribution in Figure 5(b), the jumps are due to an adjusted bin scheme. We merged the measured 99 size channels to 9 channels, to account for the different refractive indices of the particles used during calibration (Bozem et al., 2025). This rearrangement produces jumps in the size distribution as there is no smooth transition from smaller to larger particles as in the high resolved measurements with 99 channels. The range of refractive indices used during calibration (1,34 - 1,39) and the new bin scheme allows for a better representation of real atmospheric aerosol.

Thank you for the consideration of taking the measurement uncertainty as error bars. We corrected the figure with the measurement uncertainties as error bars.

Possible sources for the observed high CO values in the stratosphere are advected biomass burning residues or local uplifted pollution from the surface. We analyzed the aerosol size distributions measured by the UHSAS at the Learjet in order to find some hints for possible biomass burning in these size distributions. Therefore, we calculated the aerosol volume distribution over 10 s, which is sufficient enough to average over the full peak of aerosol number concentration. Furthermore, we also calculated volume distributions during the local UTLS background between the two consecutive pollution events and the tropospheric background in the middle of the pattern without observed pollution. The detailed times are also given in Table C1. These volume distributions (Fig. 5b) show significant differences between the UTLS

background and the polluted air masses. While the UTLS background shows a constant distribution up to 500 nm followed by a decrease in larger aerosol particles, we observe a modal distribution with a mode between 200 and 400 nm during the pollution events. This observed mode is robust against the measurement uncertainty, with only small overlaps larger than 300 nm there the uncertainty is highest due to instrumental issues of gain stitching. Such a mode in the volume distribution has previously been reported for observed aged biomass burning aerosol (Alonso-Blanco et al., 2014; Ditas et al., 2018; Brock et al., 2021; Schill et al., 2022; Holanda et al., 2023).

p. 12, last paragraph: You argue here and in the following that you have at least up to the LMS a non-negligible amount of soot in your particles (see also Fig. 7). And in the troposphere it seems to be (for me) unrealistically high. This will definitely affect your optical particle measurements, was this considered in your data processing? Otherwise you cannot trust the distributions in Fig. 5b.

We agree that the amount of soot is very high. But we also want to emphasize that this is only a rough estimation of soot and a best guess with high uncertainties. Unfortunately, there have been no soot measurements during the TPEx mission. In the processing of the UHSAS data there is no chance of reprocessing with different refractive indices, but we calibrated the UHSAS with a large spread of refractive indices in order to adjust the bin scheme and cover many types of atmospheric aerosol. This calibration is done to better trust the measured size distributions. We reformulated the paragraph of BC calculation to emphasize the uncertainties of this method and discuss the likely too high concentrations.

The estimation of the rBC mass concentration is only possible, because both instruments measure in the same size range. We also have to emphasize that this estimation is only a best guess estimation with high uncertainties, based on several assumptions, and the estimated mass concentration has to be regarded as upper limit. The measurement uncertainties of the CARBIC-AMS are already in the range of 30 % (Canagaratna et al., 2007; Bahreini et al., 2009). The conversion of the number size distribution measured by the UHSAS to a volume distribution add a further considerable uncertainty. Since we used a difference between two measurements the calculated rBC mass concentration can in principle include other components, such as sea salt or mineral dust in addition to rBC. In the remote atmosphere, mineral dust and sea spray are typically found in the coarse mode above 1 µm diameter, although some contribution to the submicron aerosol mass has also been observed (Brock et al., 2021). Furthermore, BB aerosol may also contain non-refractory salts such as KCl (Dang et al., 2022). The gained amount of rBC is too high for real atmospheric values in the order of less than 5 % (Yu et al., 2019). Therefore, this approximation can be regarded only as upper limit, but is consistent with the observed enhanced CO and the SEM particle analysis. The information on black carbon may serve as an indicator of particle origin, especially regarding biomass burning. Furthermore, the estimated rBC fraction is influenced by an instrumental drift of the CARIBIC-AMS in the beginning of the flight due to short preparation times with respect to reducing the background in the vacuum system. For the estimation we assume that all refractory aerosol that is not detected by CARIBIC-AMS is composed of black carbon.

p. 16, Fig. 8: I understand the "normalized by sample volume" on the y-axis (please remove the empty parenthesis), but I do not understand the "normalized to the flight blank and sampling time" in the figure caption. And these are two different statements, right?

Thank you for the careful reading, here was a small error in the formulation. We reformulated it to the following and adjusted the figure.

...signal of the flight blank was subtracted and the measurements were normalized according to the sampling volume...

p. 16, l. 319: The interpretation of the particulate BB tracers, what do the results indicate? That there have always been BB influence during the sampling period?

The results indicate that the tracers were confirmed as well by UHPLC-HRMS, by using authentic standards. Here, the tracers were identified using authentic standards. The results therefore indicate a slight increase of BB-tracers for F07 Filter2, though it is correct that the sampling time was not sufficient to get extremely good signals and moreover significant differences. The main issue here is the time resolution of the filters, which is not precisely able to catch a 2 min BB-event. However, we wanted to highlight the broad variety of confirming instrumentation that we had onboard, strengthening our analysis.

p. 16, l. 323: The short lifetime of levoglucosan is how long? Please provide this number in the text.

We added the lifetime of levoglucosan to the main text, which is between 0.5 and up to 4 days according to Hoffmann et al. 2009 (https://doi.org/10.1021/es902476f).

p. 19, l. 374: The whole paragraph. Either you show results of the radiative impact here or you give an estimate on how frequent such events might occur. Otherwise the value of your results is hard to estimate, i.e. are they relevant.

In this paragraph we describe the possible radiative impact based on the measurements of the vertical gradient of potential temperature.

Furthermore, during the time of the review process another publication by was submitted (Khaykin et al., 2025) which also describes the uplift of BB pollution, showing this relative new pathway of biomass burning aerosol uplift. In contrast to our observation. Khaykin et al. show the local uplift close to the fires, but they also show that this process is relevant, especially for smaller biomass burning events with not sufficient energy for self-lofting of the pollution. A radiation simulation for our observation is still planned, but this is still work in progress.

We added the discussion of the new publication in this section to show the relevance of this process:

Referring to the observations by Ditas et al. (2018) typical heating rates as consequence of rBC are in the magnitude of 0.07 K  $\cdot$  d-1 up to 0.44 K  $\cdot$  d-1 in extreme cases, which shows a

significant contribution to the radiative feedback on lower stratospheric dynamics. Combining the uncertainties of the rBC approximation and the gradient calculation of potential temperature (31 %, see Bozem et al. (2025)) we end up in the same regime of expected heating rates of roughly 0.1 K  $\cdot$  d-1 of the tropopause region after the WCB uplift.

- [...] As the transport of rBC and BB plumes was mostly studied in the presence of pyroconvection and fast uplift of pollutants to the tropopause region, current studies like this and that of Khaykin et al. (2025) show an additional transport pathway towards the UTLS. This uplift can occur close to the fire locations like in Khaykin et al. (2025) or after low-level long range transport far away from the pollution source as shown in this study.
- [...] Finally, we were able to show an additional pathway of BB pollution into the extratropical tropopause region by WCB uplift. In contrast to other studies (Khaykin et al., 2025), we observe this uplift mechanism after low-level long-range transport.

p. 20, Fig. 11: Again I doubt the BC fraction of 40% at the lowest flight level. I assume that there must have been other particle material like dust.

We agree that the fraction of BC is much too high. As mentioned earlier we reformulated the part of the BC estimation to emphasize the uncertainties of this method and that this fraction is the highest possible fraction which is possible without any other particle types.

p. 28, Fig. D1: The first too low AMS mass concentrations, could these be caused by the AMS warm- up time?

Yes, this is correct. Unfortunately, the warm-up time during this mission was rather short, compared to other measurement campaigns. Therefore, the first part of the flight the mass concentrations are lower which also increases the uncertainties in the soot calculation but can not be responsible for the whole difference. This discussion is also added to the main part of the manuscript in the paragraph where we introduce the soot estimation.

**Technical corrections:**

p. 3, l. 76: Please change to "The TOSS and the aircraft were equipped ..."

We rephrased this sentence as suggested.

- p. 3, l. 81: Please specify what is meant with "stratospheric intrusions which increase during the flight", are they reaching deeper into the troposphere or are they covering a larger area or do they occur more frequently?
- [...] which are growing in spatial extent during the flight (green patches in Fig. 1b).

p. 4, Fig. 1: The flight path is displayed in red, not in black, as stated in the figure caption.

Thank you for this note, as it was a change during submission, we corrected it in the caption.

p. 6, l. 132: Please provide a reference publication for the instrument and the uncertainties.

We added the requested reference, which is Müller et al. (2012) and Kunkel et al. (2019).

p. 7, l. 149: The first two sentences of this paragraph provide partly the same information. Please remove this doubling.

**Removed as suggested.**

p. 8, l. 208: The information that you use N2O for defining the chemical tropopause is already given in line 203. Please remove one of the two sentences.

**Removed as suggested.**

p. 9, l. 211: This subsection is quite long, do you see any chance to split it? This would make it easier for the reader to follow your line of arguments.

We divided this subsection into 2 subsections to make the reading easier.

p. 9, l. 235: The "recent particle formation event" could it be an aircraft plume encounter? Did you check for instance flightradar24 for such a possibility?

We did not check this event in detail, because it is not focus of this study. But we agree that this is an interesting feature and we can not exclude that this might be an aircraft plume encounter.

p. 10, l. 237: "the chemical stratosphere" is probably not the correct term (what would this be?), you mean "chemically stratospheric air".

You are right, we mean chemically stratospheric air and adjusted this formulation in the revised version.

p. 11, Fig. 4: Please provide the particle size range information in the legend of Fig. 4a.

We added the size information to the axis label of Fig. 4a.

p. 13, l. 285: You refer to Fig. D1 and discuss it here in the main text. Consequently, the figure should be shown here. Same for figure F2 later on.

As suggested, we moved both figures into the main part of the manuscript.

p. 17, l. 346: The half sentence "and the biomass burning pollution trajectories" does not fit here and seems to me remnant from a former test version.

Thank you for the careful reading, as you are right, we corrected this term.

p. 18, l. 359: Please change "aerosol number concentration" to "aerosol particle number concentrations".

Changed as requested.

p. 21, l. 407: "the chemical ... stratosphere" is wrong again, you mean "in chemically stratospheric air".

As written above, we rephrased this formulation.

p. 21, l. 429: Please exchange "into" with "on", otherwise the sentence does not make sense.

We changed the phrasing to the requested formulation.

p. 28, l. 477: there is a space missing in-between "gcm" and also in ".5min" in the next line.

We added the missing space in the unit.

p. 31, Fig. F1: the red line indicating the back trajectories are hard to see over the orange/brown background. Please use a different color, e.g. bright green.

We changed the color to a bright yellow.

In addition to the requested revisions, we reformulated the following paragraphs in the manuscript in order to make some statements clearer and strengthen the analysis on request of one co-author (line numbers according to the track changes document):

- In the main part of the manuscript, we replaced TOSS by TPC-TOSS to be consistent with the now included manuscript by Bozem et al. (2025).
- Line 33: lowermost stratosphere (LMS)
- Line 35: There are several additional processes which influence the chemical composition and other properties of the aerosol on shorter timescales and more locally, such as convective events, planetary and synoptic scale waves, associated with baroclinic instabilities and vertical transport from the PBL to the UT ahead the surface cold fronts by warm conveyor belts (WCBs). These processes often generate strong shear, thus favorable conditions for turbulence and mixing (Zahn et al., 2000; Brioude et al., 2007; Kaluza et al., 2021, 2022; Lachnitt et al., 2023)
- Line 137: For the simultaneous measurement of the trace gases nitrous oxide (N2O) and carbon monoxide (CO) the Quantum Cascade Laser based spectrometer University Mainz airborne QCL Spectrometer (UMAQS) is used (Müller et al., 2015; Kunkel et al., 2019)
- Line 215: The tropopause height is highly variable in time and space and depends further on the used definition. During summer months the dynamical 2 PVU tropopause tends to be lower than thermal WMO tropopause or the PV-gradient tropopause (Kunz et al., 2011; Turhal et al., 2024).
- Line 384: In contrast to the pollution trajectories, those trajectories, which indicate pristine UTLS background, are also crossing Canada and regions with active wildfires, but in higher altitudes (Fig. 9b and Fig. 10).
- Line 390: In addition to differences of the altitude of the trajectories, we observed differences between upwind velocity during the uplift process into the UTLS.

**Additional References:**

[revised manuscript text omitted]

P. Joppe et al.

**Author comments to Reviewer #4**

The reviewer comments are written in this font style and color.

Our answers are written in this font style and color.

Changes in the revised version of the manuscript are written in red.

In this work, Joppe et al. studied the transport of biomass burning aerosol into the extratropical upper troposphere and lower stratosphere (exUTLS) through a warm conveyor belt. This was a case study of one event that occurred during Flight 7 in the TropoPause compositon gradients and mixing Experiment (TPex) campaign. During the TPex campaign, a Learjet 35A flew a package of in-situ and offline measurements. The in-situ aerosol measurements were sizing from a UHSAS, OPC, and mcCPC, as well as non-refractory aerosol chemical composition from a miniAMS. The in-situ gas phase measurements were N2O, CO, and O3. There were also offline aerosol measurements. These include a filter-based collection that was rinsed and run through an HPLC and then through an orbitrap. There was also a cascade impactor that held TEM grids for offline SEM-EDX analysis. Finally, a similar suite of in-situ measurements was taken on the Towed Sensored Shuttle (TOSS) to measure vertical gradients. Overall, this is a very well-written paper with very few technical comments. However, I have two major comments that, if addressed, I think will increase the clarity of the paper and strengthen its message.

Thank you very much for the detailed reading of the manuscript and the positive consideration of the topic for publication. Thanks also for the comments which help to improve the manuscript.

**Major Comments**

1. My first major comment is on the rBC calculation. Refractory BC is generally a small component (<5%) of fresh biomass burning smoke. For example, the study by Yu et al. shows that freshly injected pyroCb smoke in the stratosphere from the Pacific Northwest Event was ~2% rBC, which accounted for its incredible rise to 23 km. The estimate that rBC may be almost 40% and 20% of the tropospheric and UTLS aerosol mass, respectively, is difficult to reconcile with many of the previous in situ measurements at these altitudes (see ATom and HIPPO campaigns). The authors are careful to point out their rBC calculation is an upper estimate, but I think these estimates are so far off that this warrants further explanation.

Thank you very much for also pointing out these uncertainties. We rephrased this paragraph, explaining the derivation of rBC in more detail, focused on all the uncertainties and compare our derived mass fractions with real measurements from other missions to clarify that this estimation should be regarded as our best guess option.

The estimation of the rBC mass concentration is only possible, because both instruments measure in the same size range. We also have to emphasize that this estimation is only a best guess estimation with high uncertainties, based on several assumptions, and the estimated mass concentration has to be regarded as upper limit. The measurement uncertainties of the CARBIC-AMS are already in the range of 30 % (Canagaratna et al., 2007; Bahreini et al., 2009). The conversion of the number size distribution measured by the UHSAS to a volume distribution add a further considerable uncertainty. Since we used a difference between two measurements the calculated rBC mass concentration can in principle include other components, such as sea salt or mineral dust in addition to rBC. In the remote atmosphere, mineral dust and sea spray are typically found in the coarse mode above 1 µm diameter, although some contribution to the submicron aerosol mass has also been observed (Brock et al., 2021). Furthermore, BB aerosol may also contain non-refractory salts such as KCl (Dang et al., 2022). The gained amount of rBC is too high for real atmospheric values in the order of less than 5 % (Yu et al., 2019). Therefore, this approximation can be regarded only as upper limit, but is consistent with the observed enhanced CO and the SEM particle analysis. The information on black carbon may serve as an indicator of particle origin, especially regarding biomass burning. Furthermore, the estimated rBC fraction is influenced by an instrumental drift of the CARIBIC-AMS in the beginning of the flight due to short preparation times with respect to reducing the background in the vacuum system. For the estimation we assume that all refractory aerosol that is not detected by CARIBIC-AMS is composed of black carbon.

2. The above major point affects the heating rate calculated in Section 3.3, and will also affect the radiation simulations cited at the end of Section 3.3.

**We included these uncertainties also in the later discussions:**

Therefore, considering these observations in the trajectory data started at the Learjet position, we hypothesize that the changes in the static stability may also be forced by radiative effects of the transported rBC in the absence of (cirrus) clouds. Referring to the observations by Ditas et al. (2018) typical heating rates as consequence of rBC are in the magnitude of 0.07 K  $\cdot$  d-1 up to 0.44 K  $\cdot$  d-1 in extreme cases, which shows a significant contribution to the radiative feedback on lower stratospheric dynamics. Combining the uncertainties of the rBC approximation and the gradient calculation of potential temperature (31 %, see Bozem et al. (2025)) we end up in the same regime of expected heating rates of roughly 0.1 K  $\cdot$  d-1 of the tropopause region after the WCB uplift.

3. The other major comment is that the SOAP analysis, in which the filters were washed and analyzed with HPLC and orbitrap mass spectrometry, seems relatively weak to me. The CO + UHSAS, AMS, TEM grid and back trajectory analysis are all much stronger evidence that this pollution event is from biomass burning. Certainly, the technique seems strong, but I feel that removing these results from the main paper would tighten its focus and make it stronger. This reviewer suggests that they can be moved to the supplemental section if needed.

We agree that for this analysis the SOAP results show only a minor support. We moved this analysis as suggested to the appendix.

**3 Minor/Technical Comments**

• P11L244: In Figure 4, it is unclear to me whether the data in all the plots is from the TOSS or from the Learjet cabin.

We rephrased the caption to point out the all the data is from the Learjet.

[...] represents different in-situ measurements onboard the Learjet:

• P12L272: How many of the grid particles were non-refractory vs refractory?

The number of particles smaller 500 nm is dominated by volatile particles. The larger particles (> 500 nm) show a fraction of at least 50 % refractory particles. An exact number cannot be given, because we do not know the losses of the volatile particles.

• P12L274: Are soot particles considered refractory or non-refractory in this analysis? More details regarding the TEM grid analysis needs to be outlined either here or in the experimental section.

Soot particles are considered as refractory in this analysis, as they are stable under the electron beam and do not dissolve. We reformulated the technical introduction of the multimini8 impactor in Section 2.2.3 to provide more information.

Additionally, during all flights UTLS particle samples were collected by the miniaturized MultiMINI8 casacade impactor unit. This self-developed Integrated Aerosol Sampling System, which is based on the former MultiMINI design (Ebert et al., 2016) was designed for the use within the wing pod of the Learjet. In total 8 two stage impactors (particle diameter: fine stage  $0.1-1~\mu m$ ; coarse stage  $>1~\mu m$ ) are integrated in this sampling unit. Particles were deposited on TEM grids, which are best suited for later offline individual particle analysis by electron microscopic methods. Size, morphology, and elemental composition of the particles were studied by scanning electron microscopy (SEM) and energy-dispersive X-ray microanalysis (EDX). SEM-EDX was carried out with a FEI ESEM Quanta 400F (Eindhoven, The Netherlands) equipped with a X max 80 energy-dispersive X-ray detector (Oxford Instruments, Abingdon, UK), which enables the analysis of elements with  $Z \ge 5$ . All investigations were carried out at 12.5~kV acceleration voltage and spot size 4 (beam diameter  $\approx 30~nm$ ). The particles were studied without coating in the high vacuum mode of the instruments ( $\approx 5 \cdot 10^{-6}~mbar$  sample

chamber pressure). Particle types were classified based on chemical composition and in case of biomass burning particles and soot additionally based on their typical morphology.

• P13L284: There needs to be a space between g and cm-3.

We added the space.

• P14L291: If you look at Figure 11 of Brock 2021, you can see that there is some contribution from sea salt and dust to the submicron aerosol mass. Furthermore, there are likely to be non-refractory salts like KCl in biomass burning aerosol.

We reformulated the paragraph, including the uncertainties with other substances and we only can give the maximum possible amount of BC in the absence of other particle types.

In the remote atmosphere, mineral dust and sea spray are typically found in the coarse mode above 1  $\mu$ m diameter, although some contribution to the submicron aerosol mass has also been observed (Brock et al., 2021). Furthermore, BB aerosol may also contain non-refractory salts such as KCl (Dang et al., 2022).

In addition to the requested revisions, we reformulated the following paragraphs in the manuscript in order to make some statements clearer and strengthen the analysis on request of one co-author (line numbers according to the track changes document):

- In the main part of the manuscript, we replaced TOSS by TPC-TOSS to be consistent with the now included manuscript by Bozem et al. (2025).
- Line 33: lowermost stratosphere (LMS)
- Line 35: There are several additional processes which influence the chemical composition and other properties of the aerosol on shorter timescales and more locally, such as convective events, planetary and synoptic scale waves, associated with baroclinic instabilities and vertical transport from the PBL to the UT ahead the surface cold fronts by warm conveyor belts (WCBs). These processes often generate strong shear, thus favorable conditions for turbulence and mixing (Zahn et al., 2000; Brioude et al., 2007; Kaluza et al., 2021, 2022; Lachnitt et al., 2023)
- Line 137: For the simultaneous measurement of the trace gases nitrous oxide (N2O) and carbon monoxide (CO) the Quantum Cascade Laser based spectrometer University Mainz airborne QCL Spectrometer (UMAQS) is used (Müller et al., 2015; Kunkel et al., 2019)
- Line 215: The tropopause height is highly variable in time and space and depends further on the used definition. During summer months the dynamical 2 PVU tropopause tends to be lower than thermal WMO tropopause or the PV-gradient tropopause (Kunz et al., 2011; Turhal et al., 2024).
- Line 384: In contrast to the pollution trajectories, those trajectories, which indicate pristine UTLS background, are also crossing Canada and regions with active wildfires, but in higher altitudes (Fig. 9b and Fig. 10).
- Line 390: In addition to differences of the altitude of the trajectories, we observed differences between upwind velocity during the uplift process into the UTLS.

**Additional References:**

Bozem, H., Joppe, P., Li, Y., Emig, N., Afchine, A., Breuninger, A., Curtius, J., Hofmann, S., Ismayil, S., Kandler, K., Kunkel, D., Kutschka, A., Lachnitt, H.-C., Petzold, A., Richter, S., Röschenthaler, T., Rolf, C., Schneider, L., Schneider, J., Vogel, A., and Hoor, P.: The TropoPause Composition TOwed Sensor Shuttle (TPC-TOSS): A new airborne dual platform approach for atmospheric composition measurements at the tropopause, EGUsphere [preprint], https://doi.org/10.5194/egusphere-2025-3175, 2025.

Breuninger, A., Joppe, P., Wilsch, J., Schwenk, C., Bozem, H., Emig, N., Merkel, L., Rossberg, R., Keber, T., Kutschka, A., Waleska, P., Hofmann, S., Richter, S., Ungeheuer, F., Dörholt, K., Hoffmann, T., Miltenberger, A., Schneider, J., Hoor, P., and Vogel, A. L.: Organic aerosols mixing across the tropopause and its implication for anthropogenic pollution of the UTLS, EGUsphere [preprint], https://doi.org/10.5194/egusphere-2025-3129, 2025.

Dang, C., Segal-Rozenhaimer, M., Che, H., Zhang, L., Formenti, P., Taylor, J., Dobracki, A., Purdue, S., Wong, P.-S., Nenes, A., Sedlacek III, A., Coe, H., Redemann, J., Zuidema, P., Howell, S., and Haywood, J.: Biomass burning and marine aerosol processing over the southeast Atlantic Ocean: a TEM single-particle analysis, Atmospheric Chemistry and Physics, 22, 9389–9412, <a href="https://doi.org/10.5194/acp-22-9389-2022">https://doi.org/10.5194/acp-22-9389-2022</a>, 2022

Hoffmann, D., Tilgner, A., Iinuma, Y., and Herrmann, H.: Atmospheric Stability of Levoglucosan: A Detailed Laboratory and Modeling Study, Environmental Science amp; Technology, 44, 694–699, https://doi.org/10.1021/es902476f, 2009

Khaykin, S., Bekki, S., Godin-Beekmann, S., Fromm, M. D., Goloub, P., Hu, Q., Josse, B., Laeng, A., Meziane, M., Peterson, D. A., Pelletier, S., and Thouret, V.: Stratospheric impact of the anomalous 2023 Canadian wildfires: the two vertical pathways of smoke, https://doi.org/10.5194/egusphere-2025-3152, 2025

Kunz, A., Konopka, P., Müller, R., and Pan, L. L.: Dynamical tropopause based on isentropic potential vorticity gradients, Journal of Geophysical Research, 116, https://doi.org/10.1029/2010jd014343, 2011

Voigt, C., Schumann, U., Minikin, A., Abdelmonem, A., Afchine, A., Borrmann, S., Boettcher, M., Buchholz, B., Bugliaro, L., Costa, A., Curtius, J., Dollner, M., Dörnbrack, A., Dreiling, V., Ebert, V., Ehrlich, A., Fix, A., Forster, L., Frank, F., Fütterer, D., Giez, A., Graf, K., Grooß, J.-U., Groß, S., Heimerl, K., Heinold, B., Hüneke, T., Järvinen, E., Jurkat, T., Kaufmann, S., Kenntner, M., Klingebiel, M., Klimach, T., Kohl, R., Krämer, M., Krisna, T. C., Luebke, A., Mayer, B., Mertes, S., Molleker, S., Petzold, A., Pfeilsticker, K., Port, M., Rapp, M., Reutter, P., Rolf, C., Rose, D., Sauer, D., Schäfler, A., Schlage, R., Schnaiter, M., Schneider, J., Spelten, N., Spichtinger, P., Stock, P., Walser, A., Weigel, R., Weinzierl, B., Wendisch, M., Werner, F., Wernli, H., Wirth, M., Zahn, A., Ziereis, H., and Zöger, M.: ML-CIRRUS: The Airborne Experiment on Natural Cirrus and Contrail Cirrus with the High-Altitude Long-Range Research Aircraft HALO, Bulletin of the American Meteorological Society, 98, 271–288, https://doi.org/10.1175/bams-d-15-00213.1, 2017

von der Weiden, S. L., Drewnick, F., and Borrmann, S.: Particle Loss Calculator – a new software tool for the assessment of the performance of aerosol inlet systems, Atmos. Meas. Tech., 2, 479-494, 10.5194/amt-2-479-2009, 2009.

Zahn, A., Brenninkmeijer, C. A. M., Maiss, M., Scharffe, D. H., Crutzen, P. J., Hermann, M., Heintzenberg, J., Wiedensohler, A., Güsten, H., Heinrich, G., Fischer, H., Cuijpers, J. W. M., and van Velthoven, P. F. J.: Identification of extratropical two-way troposphere-stratosphere mixing based on CARIBIC measurements of O3, CO, and ultrafine particles, Journal of Geophysical Research: Atmospheres, 105,1527–1535, https://doi.org/10.1029/1999jd900759, 2000

---

## Referee Report (RR1)

**Review of the revised ACP manuscript acp-2025-1346**

"Transport of Biomass Burning Aerosol into the Extratropical Tropopause Region over Europe via Warm Conveyor Belt Uplift"

By Philipp Joppe, Johannes Schneider, Jonas Wilsch, Heiko Bozem, Anna Breuninger, Joachim Curtius, Martin Ebert, Nicolas Emig, Peter Hoor, Sadath Ismayil, Konrad Kandler, Daniel Kunkel, Isabel Kurth, Hans-Christoph Lachnitt, Yun Li, Annette Miltenberger, Sarah Richter, Christian Rolf, Lisa Schneider, Cornelis Schwenk, Nicole Spelten, Alexander L. Vogel, Yafang Cheng, and Stephan Borrmann.

The revised manuscript improved, but there is still a small number of issues which are not well addressed. One of them is major an request some bigger change.

**Specific remarks:**

p. 2, l. 35: Please cite not only references from your group, there are other publications which have shown which are related to your topic, for instance Brioude et al, Atmos. Chem. Phys., 7, 4229–4235, 2007 or Zahn et al., J. Geophys. Res. 105, 1527-1535, 2000.

You state that you added the Brioude et al. reference, but it does not show up in your list of additional references. As I could not find the revised manuscript, I could not check if it is listed there.

p. 6, l. 115: You state that you operated 3 CPCs, but you provide only two lower threshold diameters. Why?

I fully understand your answer and your reasons, my request is to let the reader know you arguments (add it to the text), otherwise the reader will puzzle as I did.

p. 7, last paragraph: I'm not an expert in this, but can you exclude particle changes during the storage times? Either way, could you please write a sentence, if these can be excluded (and if yes, why) or if they are of minor importance or ...

Again, your answer is good, but please provide this information to the reader and change your text accordingly. (As I said, I did not have the revised manuscript, so sorry, if you already did this)

p. 11, Fig. 5.: This figure puzzles me a lot. First of all, why is the first a delay in the AMS data and in the next peak the AMS is ahead? Secondly the strange looking UT/LS background volume size distribution, why are there jumps of 50% in relative narrow size bins? Is there an issue with the assumed refractive index of the particles? And the error bars are misleading, the measurement period is short, hence it would be much better to indicate the measurement uncertainty here, which should be some ten percent, I guess. Volume size distributions derived from OPC measurements are highly uncertain! The data behind this figure need a deeper analysis. The different colors in Fig. 5b are not explained in the legend.

I do not understand your averaging argument concerning the shift. Right averaging does not lead to a shift. Moreover, it would not explain why one peak is delayed and one is ahead, because it would affect both peaks equally.

Concerning Fig. 5b my comment was probably not specific enough. Of course there are jumps when you merge bins, what I meant I that there is no physical reason, why there should be such very narrow particle modes in your particle volume size distribution. I have seen this before in OPC data and they were cause by the wrong refractive index/particle shape assumptions. Hence I do not trust

the shape of you volume size distributions, but admit that the relative amplitudes for the different regions could be meaningful (limitation: see below). As I assume that you do not like to carry out a more detailed optical analysis of your data, my suggestion would be to state this in the text. The shape of the size distribution (the ups and downs) are probably errors of the analysis procedure (wrong refractive index and/or shape), but the amplitudes show (somehow, see below) the relative differences between the different regions.

p. 12, last paragraph: You argue here and in the following that you have at least up to the LMS a non-negligible amount of soot in your particles (see also Fig. 7). And in the troposphere it seems to be (for me) unrealistically high. This will definitely affect your optical particle measurements, was this considered in your data processing? Otherwise you cannot trust the distributions in Fig. 5b.

The problem with the very high potential BC content was also raised by the second reviewer. I also/still believe that even the statement of an "rough estimate" in your reply is wrong, these are unrealistically high fractions and if they are not caused by the uncertainties in the two methods there was likely another aerosol material there.

But this was not the focus of my question. What does not get together is that you state on the one hand a very high BC content in your particles, but on the other hand work with OPC response functions for non-absorbing particles only (as far as I could see in the Bozem et al, 2025 manuscript). Moreover, this BC contents changes with the region under consideration, hence even the relative ratios of the magnitudes in Fig. 5b are questionable.

Sorry for saying this, but this is a big, for me unacceptable inconsistency in you data and makes the absolute values questionable. This issue must be addressed somehow.

---

## Author Response (AR2)

**ACP manuscript acp-2025-1346**

Transport of Biomass Burning Aerosol into the Extratropical Tropopause Region over Europe via Warm Conveyor Belt Uplift

P. Joppe et al.

**Author comments to Reviewer #3**

The reviewer comments are written in this font style and color.

Our answers are written in this font style and color.

Changes in the revised version of the manuscript are written in red.

Review of the revised ACP manuscript acp-2025-1346

"Transport of Biomass Burning Aerosol into the Extratropical Tropopause Region over Europe via Warm Conveyor Belt Uplift"

By Philipp Joppe, Johannes Schneider, Jonas Wilsch, Heiko Bozem, Anna Breuninger, Joachim Curtius, Martin Ebert, Nicolas Emig, Peter Hoor, Sadath Ismayil, Konrad Kandler, Daniel Kunkel, Isabel Kurth, Hans-Christoph Lachnitt, Yun Li, Annette Miltenberger, Sarah Richter, Christian Rolf, Lisa Schneider, Cornelis Schwenk, Nicole Spelten, Alexander L. Vogel, Yafang Cheng, and Stephan Borrmann.

The revised manuscript improved, but there is still a small number of issues which are not well addressed. One of them is major and request some bigger change.

**Specific remarks:**

p. 2, l. 35: Please cite not only references from your group, there are other publications which have shown which are related to your topic, for instance Brioude et al, Atmos. Chem. Phys., 7, 4229–4235, 2007 or Zahn et al., J. Geophys. Res. 105, 1527-1535, 2000.

You state that you added the Brioude et al. reference, but it does not show up in your list of additional references. As I could not find the revised manuscript, I could not check if it is listed there.

We are sorry that you did not had access to the revised version. We checked it again and it is included in the revised version. Further, we include it in the additional reference list below this author comment.

p. 6, l. 115: You state that you operated 3 CPCs, but you provide only two lower threshold diameters. Why?

I fully understand your answer and your reasons, my request is to let the reader know you arguments (add it to the text), otherwise the reader will puzzle as I did.

We added it accordingly to the new revised version.

Here, we decided to operate two of the three mc-CPC channels at the same cutoffs, to cross-check the data quality of the aerosol number concentration during the flights.

p. 7, last paragraph: I'm not an expert in this, but can you exclude particle changes during the storage times? Either way, could you please write a sentence, if these can be excluded (and if yes, why) or if they are of minor importance or ...

Again, your answer is good, but please provide this information to the reader and change your text accordingly. (As I said, I did not have the revised manuscript, so sorry, if you already did this)

It was added to the revised version, which was unfortunately not available to you.

After sampling, the filters were sealed in aluminium foil and stored in a freezing box to minimize artefacts and losses of the collected aerosol particles (Resch et al., 2023).

p. 11, Fig. 5.: This figure puzzles me a lot. First of all, why is the first a delay in the AMS data and in the next peak the AMS is ahead? Secondly the strange looking UT/LS background volume size distribution, why are there jumps of 50% in relative narrow size bins? Is there an issue with the assumed refractive index of the particles? And the error bars are misleading, the measurement period is short, hence it would be much better to indicate the measurement uncertainty here, which should be some ten percent, I guess. Volume size distributions derived from OPC measurements are highly uncertain! The data behind this figure need a deeper analysis. The different colors in Fig. 5b are not explained in the legend.

I do not understand your averaging argument concerning the shift. Right averaging does not lead to a shift. Moreover, it would not explain why one peak is delayed and one is ahead, because it would affect both peaks equally.

We checked this data again carefully. These shifts seem to come from reduced instrument sensitivity due to short preparation time and ongoing background reduction. Therefore, the first peak is delayed. The second peak in our opinion is not ahead and is going more or less in parallel with the UHSAS increase. Nevertheless, we removed the AMS concentration from this figure to reduce the complexity and avoid possible misinterpretation.

Concerning Fig. 5b my comment was probably not specific enough. Of course there are jumps when you merge bins, what I meant I that there is no physical reason, why there should be such very narrow particle modes in your particle volume size distribution. I have seen this before in OPC data and they were cause by the wrong refractive index/particle shape assumptions. Hence I do not trust the shape of you volume size distributions, but admit that the relative amplitudes for the different regions could be meaningful (limitation: see below). As I assume that you do not like to carry out a more detailed optical analysis of your data, my suggestion would be to state this in the text. The shape of the size distribution (the ups and downs) are probably errors of the analysis procedure (wrong refractive index and/or shape), but the amplitudes show (somehow, see below) the relative differences between the different regions.

We thank you for the clarification about your comment and are sorry for the not satisfying answer to it. You are right that there is a huge impact of the refractive index on the size distribution measurements. We tried our best with calibrations and adjusted bin schemes to reduce these uncertainties. However, there are still remaining limitations and uncertainties. To account for those, we changed this Figure to show the uncertainty of the distributions due to undersizing as consequence of absorption processes of the aerosol particles. For this, we used the results published by Moore et al., 2021. They provide sizing uncertainties of the UHSAS with respect to different refractive indices and potential absorption by black carbon. We added these uncertainties to our measured size distributions and show the uncertainty range. As a result, the regions can still be differentiated from each other.

As optical particle detection reveals some uncertainties due to absorbing aerosol particles, especially when calculating volume distributions, we additionally provide an uncertainty range. For this, we use the sizing uncertainties for a UHSAS according to Moore et al. (2021). Based on in-situ observations and laboratory studies, they provide sizing errors for different particle types, especially for wildfire biomass burning aerosol. Following Moore et al. (2021), we calculated the uncertainty range for potential 3 % oversizing up to 20 % undersizing as consequence of light absorption by wildfire biomass burning aerosol particles. The inferred volume distributions (Fig. 5b) show significant differences between the UTLS background and the polluted air masses. Even though the uncertainty range is high, we observe a modal distribution with a mode between 200 and 400 nm during the pollution events. This mode is robust against the measurement uncertainty and can be differentiated from the UTLS and tropospheric background.

p. 12, last paragraph: You argue here and in the following that you have at least up to the LMS a non-negligible amount of soot in your particles (see also Fig. 7). And in the troposphere it seems to be (for me) unrealistically high. This will definitely affect your optical particle measurements, was this considered in your data processing? Otherwise you cannot trust the distributions in Fig. 5b.

The problem with the very high potential BC content was also raised by the second reviewer. I also/still believe that even the statement of an "rough estimate" in your reply is wrong, these are

unrealistically high fractions and if they are not caused by the uncertainties in the two methods there was likely another aerosol material there. But this was not the focus of my question. What does not get together is that you state on the one hand a very high BC content in your particles, but on the other hand work with OPC response functions for non-absorbing particles only (as far as I could see in the Bozem et al, 2025 manuscript). Moreover, this BC contents changes with the region under consideration, hence even the relative ratios of the magnitudes in Fig. 5b are questionable.

Sorry for saying this, but this is a big, for me unacceptable inconsistency in you data and makes the absolute values questionable. This issue must be addressed somehow.

We understand your concerns. As mentioned above, we now included the uncertainty range of the size distribution measurements due to a possible undersizing of absorbing particles. Since we can not exclude other aerosol types than black carbon completely, we changed the terms "rough estimate" or "upper limit" of BC to "refractory aerosol, potentially BC".

After these indications for potential BB influence within the tropopause region, we use the CARIBIC-AMS measurements of non-refractory compounds to estimate the amount of refractory aerosol (rA) which can also consists partly of black carbon (BC).

**Additional References:**

Brioude, J., Cooper, O. R., Trainer, M., Ryerson, T. B., Holloway, J. S., Baynard, T., Peischl, J., Warneke, C., Neuman, J. A., De Gouw, J., Stohl, A., Eckhardt, S., Frost, G. J., McKeen, S. A., Hsie, E.-Y., Fehsenfeld, F. C., and Nédélec, P.: Mixing between a stratospheric intrusion and a biomass burning plume, Atmospheric Chemistry and Physics, 7, 4229–4235, https://doi.org/10.5194/acp-7-4229-2007, 2007.

Moore, R. H., Wiggins, E. B., Ahern, A. T., Zimmerman, S., Montgomery, L., Campuzano Jost, P., Robinson, C. E., Ziemba, L. D., Winstead, E. L., Anderson, B. E., Brock, C. A., Brown, M. D., Chen, G., Crosbie, E. C., Guo, H., Jimenez, J. L., Jordan, C. E., Lyu, M., Nault, B. A., Rothfuss, N. E., Sanchez, K. J., Schueneman, M., Shingler, T. J., Shook, M. A., Thornhill, K. L., Wagner, N. L., and Wang, J.: Sizing response of the Ultra-High Sensitivity Aerosol Spectrometer (UHSAS) and Laser Aerosol Spectrometer (LAS) to changes in submicron aerosol composition and refractive index, Atmospheric Measurement Techniques, 14, 4517–4542, <a href="https://doi.org/10.5194/amt-14-4517-2021">https://doi.org/10.5194/amt-14-4517-2021</a>, 2021.

Zahn, A., Brenninkmeijer, C. A. M., Maiss, M., Scharffe, D. H., Crutzen, P. J., Hermann, M., Heintzenberg, J., Wiedensohler, A., Güsten, H., Heinrich, G., Fischer, H., Cuijpers, J. W. M., and van Velthoven, P. F. J.: Identification of extratropical two-way troposphere-stratosphere mixing based on CARIBIC measurements of O3, CO, and ultrafine particles, Journal of Geophysical Research: Atmospheres, 105, 1527–1535, https://doi.org/10.1029/1999jd900759, 2000.

---

## Author Response (AR3)

**Reply to technical corrections by the Editor**

Dear Dr. Philipp Joppe,

I am pleased to say that your manuscript can be accepted for publication in ACP subject to technical corrections. Please follow the ACP's author guidelines (https://www.atmospheric-chemistry-and-physics.net/policies/guidelines\_for\_authors.html) to revise the abstract of your manuscript. Specifically, the abstract of your manuscript needs to be more concise (fewer than 250 words) and the importance and implications of your results should be stated in the end of the abstract. Sincerely,

Dear Jianzhong Ma,

Jianzhong Ma

thank you for taking care of the manuscript, leading the review process and the consideration for publication. Please find below our new formulated abstract for the final version of the manuscript.

Thank you very much for taking care and leading the review process.

Sincerely,

Philipp Joppe

We present measurements from the aircraft-based TPEx (Tropopause composition gradients and mixing Experiment) mission in June 2024 over Europe. The measurement platform, a Learjet 35A, was equipped with in-situ trace gas and aerosol measurements and filter samplers for offline analysis. For vertical gradient measurements of trace species and aerosol, we conducted redundant measurements on a fully automated towed sensor shuttle (TOSS) 200 m below the aircraft. On 17 June 2024, we observed a filament with elevated aerosol number concentrations of up to 800 particles per cm3 between 100 nm and 1  $\mu$ m. This is higher by a factor of two to four than the local background. Carbon monoxide (CO) mixing ratios were larger than 100 ppbv. Single particle analysis of impactor samples using electron microscopy show characteristic biomass burning (BB) aerosol in the tropopause region. The TOSS measurements also allow the calculation of the potential temperature gradient ( $\Delta\theta \cdot \Delta z$ -1). Within the polluted filament, we observe changes towards smaller gradients, which is presumably due to an increase of potential temperature at lower altitudes by radiative heating as a consequence of the transported BB aerosol.

Trajectory analysis show air mass origin over Canada with low-level long-range transport and subsequent uplift by a warm conveyor belt (WCB) over Europe as additional pathway of pollution into the UTLS. Furthermore, this analysis yields that BB aerosol can be transported in a WCB into the UTLS there it can be mixed with stratospheric air masses.